# ACTIVATION-LEVEL UNCERTAINTY IN DEEP NEURAL NETWORKS

**Pablo Morales-Álvarez**[*]
Department of Computer Science and AI
University of Granada, Spain
`pablomorales@decsai.ugr.es`

**Daniel Hernández-Lobato**
Department of Computer Science
Universidad Autónoma de Madrid, Spain

**Rafael Molina**
Department of Computer Science and AI
University of Granada, Spain

**José Miguel Hernández-Lobato**
Department of Engineering
University of Cambridge, UK
Alan Turing Institute, London, UK

## ABSTRACT

Current approaches for uncertainty estimation in deep learning often produce too confident results. Bayesian Neural Networks (BNNs) model uncertainty in the space of weights, which is usually high-dimensional and limits the quality of variational approximations. The more recent functional BNNs (fBNNs) address this only partially because, although the prior is specified in the space of functions, the posterior approximation is still defined in terms of stochastic weights. In this work we propose to move uncertainty from the weights (which are deterministic) to the activation function. Specifically, the activations are modelled with simple 1D Gaussian Processes (GP), for which a triangular kernel inspired by the ReLu non-linearity is explored. Our experiments show that activation-level stochasticity provides more reliable uncertainty estimates than BNN and fBNN, whereas it performs competitively in standard prediction tasks. We also study the connection with deep GPs, both theoretically and empirically. More precisely, we show that activation-level uncertainty requires fewer inducing points and is better suited for deep architectures.

## 1 INTRODUCTION

Deep Neural Networks (DNNs) have achieved state-of-the-art performance in many different tasks, such as speech recognition (Hinton et al., 2012), natural language processing (Mikolov et al., 2013) or computer vision (Krizhevsky et al., 2012). In spite of their predictive power, DNNs are limited in terms of uncertainty estimation. This has been a classical concern in the field (MacKay, 1992; Hinton & Van Camp, 1993; Barber & Bishop, 1998), which has attracted a lot of attention in the last years (Lakshminarayanan et al., 2017; Guo et al., 2017; Sun et al., 2019; Wenzel et al., 2020). Indeed, this ability to "know what is not known" is essential for critical applications such as medical diagnosis (Esteva et al., 2017; Mobiny et al., 2019) or autonomous driving (Kendall & Gal, 2017; Gal, 2016).

Bayesian Neural Networks (BNNs) address this problem through a Bayesian treatment of the network weights[1] (MacKay, 1992; Neal, 1995). This will be refered to as *weight-space* stochasticity. However, dealing with uncertainty in weight space is challenging, since it contains many symmetries and is highly dimensional (Wenzel et al., 2020; Sun et al., 2019; Snoek et al., 2019; Fort et al., 2019). Here we focus on two specific limitations. First, it has been recently shown that BNNs with well-established inference methods such as Bayes by Backprop (BBP) (Blundell et al., 2015) and MC-Dropout (Gal & Ghahramani, 2016) underestimate the predictive uncertainty for instances located in-between two clusters of training points (Foong et al., 2020; 2019; Yao et al., 2019). Second, the weight-space prior does not allow BNNs to guide extrapolation to out-of-distribution (OOD) data (Sun et al., 2019; Nguyen et al., 2015; Ren et al., 2019). Both aspects are illustrated graphically in Figure 3, more details in Section 3.1.

---

[*]Work developed mostly while visiting Cambridge University, UK.

[1]The bias term will be absorbed within the weights throughout the work.

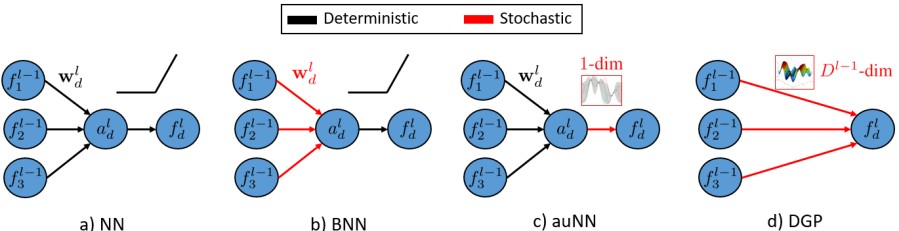

Figure 1: Graphical representation of the artificial neurons for closely related methods. The subscript $d$ and the superscript $l$ refer to the $d$-th unit in the $l$-th layer, respectively. **(a)** In standard Neural Networks (NN), both the weights and the activation function are deterministic. **(b)** In Bayesian NNs, weights are stochastic and the activation is deterministic. **(c)** In auNN (this work), weights are deterministic and the activation is stochastic. **(d)** Deep GPs do not have a linear projection through weights, and the output is modelled directly with a GP defined on the $D^{l-1}$-dimensional input space.

As an alternative to standard BNNs, Functional Bayesian Neural Nets (fBNN) specify the prior and perform inference directly in function space (Sun et al., 2019). This provides a mechanism to guide the extrapolation in OOD data, e.g. predictions can be encouraged to revert to the prior in regions of no observed data. However, the posterior stochastic process is still defined by a factorized Gaussian on the network weights (i.e. as in BBP), see (Sun et al., 2019, Sect. 3.1). We will show that this makes fBNN inherit the problem of underestimating the predictive uncertainty for in-between data.

In this work, we adopt a different approach by moving stochasticity from the weights to the activation function, see Figure 1. This will be referred to as auNN (activation-level uncertainty for Neural Networks). The activation functions are modelled with (one-dimensional) GP priors, for which a triangular kernel inspired by the ReLu non-linearity (Nair & Hinton, 2010; Glorot et al., 2011) is used. Since non-linearities are typically simple functions (e.g. ReLu, sigmoid, tanh), our GPs are sparsified with few inducing points. The network weights are deterministic parameters which are estimated to maximize the marginal likelihood of the model. The motivation behind auNN is to avoid inference in the complex space of weights. We hypothesise that it could be enough to introduce stochasticity in the activation functions that follow the linear projections to provide sensible uncertainty estimations.

We show that auNN obtains well-calibrated estimations for in-between data, and its prior allows to guide the extrapolation to OOD data by reverting to the empirical mean. This will be visualized in a simple 1D example (Figure 3 and Table 1). Moreover, auNN obtains competitive performance in standard benchmarks, is scalable (datasets of up to ten millions training points are used), and can be readily used for classification. The use of GPs for the activations establishes an interesting connection with deep GPs (DGPs) (Damianou & Lawrence, 2013; Salimbeni & Deisenroth, 2017). The main difference is the linear projection before the GP, recall Figure 1(c-d). This allows auNN units to model simpler mappings between layers, which are defined along one direction of the input space, similarly to neural networks. However, DGP units model more complex mappings defined on the whole input space, see also Figure 2a. We will show that auNN units require fewer inducing points and are better suited for deep architectures, achieving superior performance. Also, a thorough discussion on additional related work will be provided in Section 4.

In summary, the main contributions of this paper are: (1) a new approach to model uncertainty in DNNs, based on deterministic weights and simple stochastic non-linearities (in principle, not necessarily modelled by GPs); (2) the specific use of non-parametric GPs as a prior, including the triangular kernel inspired by the ReLu; (3) auNN addresses a well-known limitation of BNNs and fBNNs (uncertainty underestimation for in-between data), can guide the extrapolation to OOD data by reverting to the empirical mean, and is competitive in standard prediction tasks; (4) auNN units require fewer inducing points and are better suited for deep architectures than DGP ones, achieving superior performance.

## 2 PROBABILISTIC MODEL AND INFERENCE

**Model specification.** We focus on a supervised task (e.g. regression or classification) with training data[2] $\{\mathbf{x}_{n,:}, \mathbf{y}_{n,:}\}_{n=1}^N$. The graphical model in Figure 2b will be useful throughout this section. We

---

[2]The output is represented as a vector since all the derivations apply for the multi-output case.

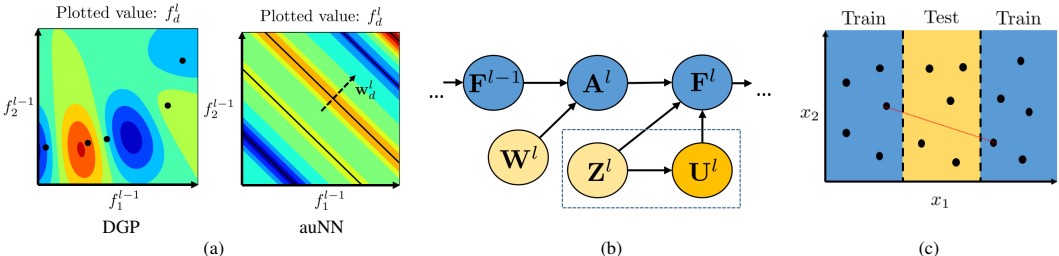

Figure 2: **(a)** Type of mappings modelled by DGP and auNN units (colours represent different values). Whereas DGP units describe complex functions defined on the whole $D^{l-1}$ dimensional input space, the linear projection through $\mathbf{w}_d^l$ in auNN yields simpler functions defined on just one direction. This is closer in spirit to NNs, requires fewer inducing points, and is better suited for deep architectures. The inducing points are shown in black (for auNN, these correspond to (hyper)planes in the input space before the projection). **(b)** Probabilistic graphical model for an auNN layer. Yellow variables are to be estimated (light ones through point estimates and the dark one through a posterior distribution). The box highlights the auxiliary variables (inducing points and their values). **(c)** Graphical representation of the UCI gap splits. In red, a segment that crosses the gap joining two training points from different components, which will be used in the experiments.

assume a model of $L$ layers, each one with $D^l$ units as in Figure 1c. Each activation is modelled with a (1D) GP prior, i.e. $f_d^l(a_d^l) \sim \mathcal{GP}(\mu_d^l, k_d^l)$, with $\mu_d^l : \mathbb{R} \to \mathbb{R}$ and $k_d^l : \mathbb{R} \times \mathbb{R} \to \mathbb{R}$. The GP hyperparameters $\boldsymbol{\theta}_d^l$ will be omitted for clarity (for the kernels used here, $\boldsymbol{\theta}_d^l$ includes the amplitude and the lengthscale). Assuming independence between units, each layer depends on the previous one as:

$$p(\mathbf{F}^l|\mathbf{F}^{l-1}, \mathbf{W}^l) = p(\mathbf{F}^l|\mathbf{A}^l) = \prod_{d=1}^{D^l} p(\mathbf{f}_d^l|\mathbf{a}_d^l), \tag{1}$$

where $\mathbf{F}^l$ is the $N \times D^l$ matrix of outputs of the $l$-th layer for $N$ inputs, $\mathbf{W}^l$ is the $D^{l-1} \times D^l$ matrix of weights in that layer, and $\mathbf{A}^l$ is the $N \times D^l$ matrix of pre-activations, i.e. $\mathbf{A}^l = \mathbf{F}^{l-1} \cdot \mathbf{W}^l$. As usual, the columns and rows of $\mathbf{F}^l$ are denoted as $\mathbf{f}_d^l$ and $\mathbf{f}_{n,:}^l$, respectively (and analogously for the other matrices). Since the activation is defined by a GP, we have $p(\mathbf{f}_d^l|\mathbf{a}_d^l) = \mathcal{N}(\mathbf{f}_d^l|\boldsymbol{\mu}_d^l, \mathbf{K}_d^l)$, with $\boldsymbol{\mu}_d^l$ (resp. $\mathbf{K}_d^l$) the result of evaluating $\mu_d^l$ (resp. $k_d^l$) on $\mathbf{a}_d^l$ (that is, $\boldsymbol{\mu}_d^l$ is a $N$-dimensional vector and $\mathbf{K}_d^l$ is a $N \times N$ matrix). To fully specify the model, the output $\mathbf{Y}$ is defined from the last layer with a distribution that factorizes across data points, i.e. $p(\mathbf{Y}|\mathbf{F}^L) = \prod_{n=1}^{N} p(\mathbf{y}_{n,:}|\mathbf{f}_{n,:}^L)$. This formulation resembles that of DGPs (Damianou & Lawrence, 2013; Salimbeni & Deisenroth, 2017). The main difference is that we model $\mathbf{F}^l|\mathbf{F}^{l-1}$ through $D^l$ 1D GPs evaluated on the pre-activations $\mathbf{A}^l$ (i.e. the projections of $\mathbf{F}^{l-1}$ through $\mathbf{W}^l$), whereas DGPs use $D^l$ GPs of dimension $D^{l-1}$ evaluated directly on $\mathbf{F}^{l-1}$, recall Figure 1(c-d).

**Variational Inference**. Inference in the proposed model is intractable. To address this, we follow standard sparse variational GP approaches (Titsias, 2009; Hensman et al., 2013; 2015), similarly to the Doubly Stochastic Variational Inference (DSVI) for DGPs (Salimbeni & Deisenroth, 2017). Specifically, in each unit of each layer we introduce $M^l$ inducing values $\mathbf{u}_d^l$, which are the result of evaluating the GP on the one-dimensional inducing points $\mathbf{z}_d^l$. We naturally write $\mathbf{U}^l$ and $\mathbf{Z}^l$ for the corresponding $M^l \times D^l$ matrices associated to the $l$-th layer, respectively. Following eq. (1), the augmented model for one layer is

$$p(\mathbf{F}^l, \mathbf{U}^l|\mathbf{F}^{l-1}, \mathbf{W}^l, \mathbf{Z}^l) = p(\mathbf{F}^l|\mathbf{U}^l, \mathbf{A}^l, \mathbf{Z}^l)p(\mathbf{U}^l|\mathbf{Z}^l) = \prod_{d=1}^{D^l} p(\mathbf{f}_d^l|\mathbf{u}_d^l, \mathbf{a}_d^l, \mathbf{z}_d^l)p(\mathbf{u}_d^l|\mathbf{z}_d^l). \tag{2}$$

Variational inference (VI) involves the approximation of the true posterior $p(\{\mathbf{F}^l, \mathbf{U}^l\}_l|\mathbf{Y})$. Following (Hensman et al., 2013; Salimbeni & Deisenroth, 2017), we propose a posterior given by $p(\mathbf{F}|\mathbf{U})$ and a parametric Gaussian on $\mathbf{U}$:

$$q(\{\mathbf{F}^l, \mathbf{U}^l\}_l) = \prod_{l=1}^{L} p(\mathbf{F}^l|\mathbf{U}^l, \mathbf{A}^l, \mathbf{Z}^l)q(\mathbf{U}^l) = \prod_{l=1}^{L}\prod_{d=1}^{D^l} p(\mathbf{f}_d^l|\mathbf{u}_d^l, \mathbf{a}_d^l, \mathbf{z}_d^l)q(\mathbf{u}_d^l), \tag{3}$$

where $q(\mathbf{u}_d^l) = \mathcal{N}(\mathbf{u}_d^l|\mathbf{m}_d^l, \mathbf{S}_d^l)$, with $\mathbf{m}_d^l \in \mathbb{R}^{M^l}$ and $\mathbf{S}_d^l \in \mathbb{R}^{M^l \times M^l}$ variational parameters to be estimated. Minimizing the KL divergence between $q(\{\mathbf{F}^l, \mathbf{U}^l\}_l)$ and the true posterior is equivalent to maximizing the following evidence lower bound (ELBO):

$$\log p(\mathbf{Y}|\{\mathbf{W}^l, \mathbf{Z}^l\}_l) \geq \text{ELBO} = \sum_{n=1}^{N} \mathbb{E}_{q(\mathbf{f}_{n,:}^L)}\left[\log p(\mathbf{y}_{n,:}|\mathbf{f}_{n,:}^L)\right] - \sum_{l=1}^{L}\sum_{d=1}^{D^l} \text{KL}\left(q(\mathbf{u}_d^l)||p(\mathbf{u}_d^l)\right). \tag{4}$$

In the ELBO, the KL term can be computed in closed-form, as both $q(\mathbf{u}_d^l)$ and $p(\mathbf{u}_d^l)$ are Gaussians. The log likelihood term can be approximated by sampling from the marginal posterior $q(\mathbf{f}_{n,:}^L)$, which can be done efficiently through univariate Gaussians as in (Salimbeni & Deisenroth, 2017). Specifically, $\mathbf{U}^l$ can be analytically marginalized in eq. (3), which yields $q(\{\mathbf{F}^l\}^l) = \prod_l q(\mathbf{F}^l | \mathbf{F}^{l-1}, \mathbf{W}^l) = \prod_{l,d} \mathcal{N}(\mathbf{f}_d^l | \tilde{\boldsymbol{\mu}}_d^l, \tilde{\boldsymbol{\Sigma}}_d^l)$, with:

$$[\tilde{\boldsymbol{\mu}}_d^l]_i = \mu_d^l(a_{id}^l) + \boldsymbol{\alpha}_d^l(a_{id}^l)^\intercal (\mathbf{m}_d^l - \mu_d^l(\mathbf{z}_d^l)), \tag{5}$$

$$[\tilde{\boldsymbol{\Sigma}}_d^l]_{ij} = k_d^l(a_{id}^l, a_{jd}^l) - \boldsymbol{\alpha}_d^l(a_{id}^l)^\intercal (k_d^l(\mathbf{z}_d^l) - \mathbf{S}_d^l) \boldsymbol{\alpha}_d^l(a_{jd}^l), \tag{6}$$

where $\boldsymbol{\alpha}_d^l(x) = k_d^l(x, \mathbf{z}_d^l)[k_d^l(\mathbf{z}_d^l)]^{-1}$ and $\mathbf{a}_{n,:}^l = \mathbf{W}^l \mathbf{f}_{n,:}^{l-1}$. Importantly, the marginal posterior $q(\mathbf{f}_{n,:}^l)$ is a Gaussian that depends only on $\mathbf{a}_{n,:}^l$, which in turn only depends on $q(\mathbf{f}_{n,:}^{l-1})$. Therefore, sampling from $\mathbf{f}_{n,:}^l$ is straightforward using the reparametrization trick (Kingma & Welling, 2013):

$$f_{nd}^l = [\tilde{\boldsymbol{\mu}}_d^l]_n + \varepsilon \cdot [\tilde{\boldsymbol{\Sigma}}_d^l]_{nn}^{1/2}, \quad \text{with } \varepsilon \sim \mathcal{N}(0, 1), \quad \text{and } \mathbf{f}_{n,:}^0 = \mathbf{x}_{n,:}. \tag{7}$$

Training consists in maximizing the ELBO, eq. (4), w.r.t. variational parameters $\{\mathbf{m}_d^l, \mathbf{S}_d^l\}$, inducing points $\{\mathbf{z}_d^l\}$, and model parameters (i.e. weights $\{\mathbf{w}_d^l\}$ and kernel parameters $\{\boldsymbol{\theta}_d^l\}$). This can be done in batches, allowing for scalability to very large datasets. The complexity to evaluate the ELBO is $\mathcal{O}(NM^2(D^1 + \cdots + D^L))$, the same as DGPs with DSVI (Salimbeni & Deisenroth, 2017).[3]

**Predictions.** Given a new $\mathbf{x}_{*,:}$, we want to compute[4] $p(\mathbf{f}_{*,:}^L | \mathbf{X}, \mathbf{Y}) \approx \mathbb{E}_{q(\{\mathbf{U}^l\})} \left[ p(\mathbf{f}_{*,:}^L | \{\mathbf{U}^l\}) \right]$. As in (Salimbeni & Deisenroth, 2017), this can be approximated by sampling $S$ values up to the $(L-1)$-th layer with the same eq. (7), but starting with $\mathbf{x}_{*,:}$. Then, $p(\mathbf{f}_{*,:}^L | \mathbf{X}, \mathbf{Y})$ is given by the mixture of the $S$ Gaussians distributions obtained from eqs. (5)-(6).

**Triangular kernel.** One of the most popular kernels in GPs is the RBF (Williams & Rasmussen, 2006), which produces very smooth functions. However, the ReLu non-linearity led to a general boost in performance in DNNs (Nair & Hinton, 2010; Glorot et al., 2011), and we aim to model similar activations. Therefore, we introduce the use of the *triangular* (TRI) kernel. Just like RBF, TRI is an isotropic kernel, i.e. it depends on the distance between the inputs, $k(x, y) = \gamma \cdot g(|x - y|/\ell)$, with $\gamma$ and $\ell$ the amplitude and lengthscale. For RBF, $g(t) = e^{-t^2/2}$. For TRI, $g(t) = \max(1 - t, 0)$. This is a valid kernel (Williams & Rasmussen, 2006, Section 4.2.1). Similarly to the ReLu, the functions modelled by TRI are piecewise linear, see Figure 6a in the main text and Figure 8 in Appendix C.

**Comparison with DGP.** The difference between auNN and DGP units is graphically illustrated in Figure 2a. Whereas DGP mappings from one layer to the next are complex functions defined on $D^{l-1}$ dimensions ($D^{l-1} = 2$ in the figure), auNN mappings are defined just along one direction via the weight projection. This is closer in spirit to NNs, whose mappings are also simpler and better suited for feature extraction and learning more abstract concepts. Moreover, since the GP is defined on a 1D space, auNN requires fewer inducing points than DGP (which, intuitively, can be interpreted as inducing (hyper)planes in the $D^{l-1}$-dimensional space before the projection).

## 3 EXPERIMENTS

In this section, auNN is compared to BNN, fBNN (Sun et al., 2019) and DSVI DGP (Salimbeni & Deisenroth, 2017). BNNs are trained with BBP (Blundell et al., 2015), since auNN also leverages a simple VI-based inference approach. In each section we will highlight the most relevant experimental aspects, and all the details can be found in Appendix B. In the sequel, NLL stands for Negative Log Likelihood. Anonymized code for auNN is provided in the supplementary material, along with a script to run it for the 1D illustrative example of Section 3.1.

### 3.1 AN ILLUSTRATIVE EXAMPLE

Here we illustrate the two aspects that were highlighted in the introduction: the underestimation of predictive uncertainty for instances located in-between two clusters of training points and the

---

[3]As in (Salimbeni & Deisenroth, 2017), there exists also a cubic term $\mathcal{O}(M^3(D^1 + \cdots + D^L))$ that is dominated by the former (since the batch size $N$ is typically larger than $M$). Moreover, in auNN we have the multiplication by weights, with complexity $\mathcal{O}(ND^{l-1}D^l)$ for each layer. This is also dominated by the former.

[4]The distribution $p(\mathbf{y}_{*,:}^L | \mathbf{X}, \mathbf{Y})$ is obtained as the expectation of the likelihood over $p(\mathbf{f}_{*,:}^L | \mathbf{X}, \mathbf{Y})$. A Gaussian likelihood is used for regression, and the Robust-Max (Hernández-Lobato et al., 2011) for classification.

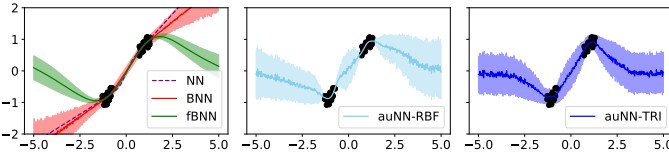

Figure 3: Predictive distribution (mean and one standard deviation) after training on a 1D dataset with two clusters of points. This simple example illustrates the main limitations of NN, BNN and fBNN, which are overcome by the novel auNN. See Table 1 for a summary and the text for details.

Table 1: Visual overview of conclusions from the 1D experiment in Figure 3. This shows that NN, BNN, fBNN and auNN increasingly expand their capabilities.

| | Epistemic uncertainty | Reverts to the mean | In-between uncertainty |
|---|---|---|---|
| NN | ✗ | ✗ | ✗ |
| BNN | ✓ | ✗ | ✗ |
| fBNN | ✓ | ✓ | ✗ |
| auNN | ✓ | ✓ | ✓ |

extrapolation to OOD data. Figure 3 shows the predictive distribution of NN, BNN, fBNN and auNN (with RBF and TRI kernels) after training on a simple 1D dataset with two clusters of points. All the methods have one hidden layer with 25 units, and 5 inducing points are used for auNN.

In Figure 3, the deterministic nature of NNs prevents them from providing epistemic uncertainty (i.e. the one originating from the model (Kendall & Gal, 2017)). Moreover, there is no prior to guide the extrapolation to OOD data. BNNs provide epistemic uncertainty. However, the prior in the complex space of weights does not allow for guiding the extrapolation to OOD data (e.g. by reverting to the empirical mean). Moreover, note that BNNs underestimate the predictive uncertainty in the region between the two clusters, where there is no observed data (this region is usually called the *gap*). More specifically, as shown in (Foong et al., 2020), the predictive uncertainty for data points in the gap is limited by that on the extremes. By specifying the prior in function space, fBNN can induce properties in the output, such as reverting to the empirical mean for OOD data through a zero-mean GP prior. However, the underestimation of in-between uncertainty persists, since the posterior stochastic process for fBNN is based on a weight-space factorized Gaussian (as BNN with BBP), see (Sun et al., 2019, Section 3.1) for details. Finally, auNN (either with RBF or TRI kernel) addresses both aspects through the novel activation-level modelling of uncertainty, which utilizes a zero-mean GP prior for the activations. Table 1 summarizes the main characteristics of each method. Next, a more comprehensive experiment with deeper architectures and more complex multidimensional datasets is provided.

## 3.2 UCI REGRESSION DATASETS WITH GAP SPLITS

Standard splits are not appropriate to evaluate the quality of uncertainty estimates for in-between data, since both train and test sets may cover the space equally. This motivated the introduction of gap splits (Foong et al., 2019). Namely, a set with $D$ dimensions admits $D$ such train-test partitions by considering each dimension, sorting the points according to its value, and selecting the middle 1/3 for test (and the outer 2/3 for training), see Figure 2c. With these partitions, overconfident predictions for data points in the gap manifest as very high values of test negative log likelihood.

Using the gap splits, it was recently shown that BNNs yield overconfident predictions for in-between data (Foong et al., 2019). The authors highlight the case of Energy and Naval datasets, where BNNs fail catastrophically. Figure 4a reproduces these results for BNNs and checks that fBNNs also obtain overconfident predictions, as theoretically expected. However, notice that activation-level stochasticity performs better, specially through the triangular kernel, which dramatically improves the results (see the plot scale). Figure 4b confirms that the difference is due to the underestimation of uncertainty, since the predictive performance in terms of RMSE is on a similar scale for all the methods. In all cases, $D = 50$ hidden units are used, and auNN uses $M = 10$ inducing points.

To further understand the intuition behind the different results, Figure 5 shows the predictive distribution over a segment that crosses the gap, recall Figure 2c. We observe that activation-level approaches obtain more sensitive (less confident) uncertainties in the gap, where there is no observed data. For instance, BNN and fBNN predictions in Naval are unjustifiably overconfident, since the output in that dataset ranges from 0.95 to 1. Also, to illustrate the internal mechanism of auNN, Figure 6a shows one example of the activations learned when using each kernel. Although it is just one example, it allows for visualising the different nature: smoother for RBF and piecewise linear for TRI. All the activations for a particular network and for both kernels are shown in Appendix C (Figure 8).

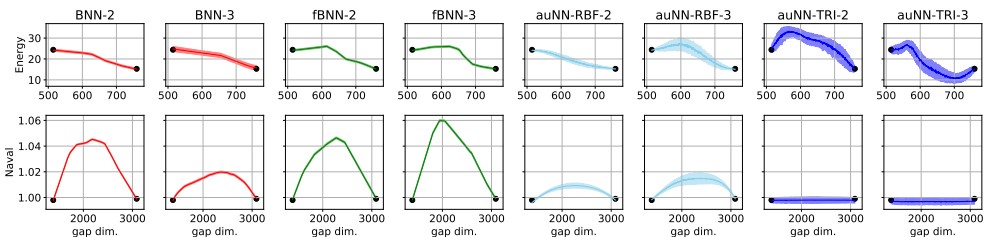

(a)                                    (b)

Figure 4: Test NLL (a) and RMSE (b) for the gap splits in Energy and Naval datasets (mean and one standard error, the lower the better). Activation-level uncertainty, specially through the triangular kernel, avoids the dramatic failure of BNN and fBNN in terms of NLL (see the scale). The similar values in RMSE reveal that this failure actually comes from an extremely overconfident estimation by BNN and fBNN, see also Figure 5.

Figure 5: Predictive distribution (mean and one standard deviation) over a segment that crosses the gap, joining two training points from different connected components. auNN avoids overconfident predictions by allocating more uncertainty in the gap, where there is no observed data.

In addition to the paradigmatic cases of Energy and Naval illustrated here, four more datasets are included in Appendix C. Figure 7 there is analogous to Figure 4 here, and Tables 4 and 5 there show the full numeric results and ranks. We observe that auNN, specially through the triangular kernel, obtains the best results and does not fail catastrophically in any dataset (unlike BNN and fBNN, which do in Energy and Naval). Finally, the performance on the gap splits is complemented by that on standard splits, see Tables 6 and 7 in Appendix C. This shows that, in addition to the enhanced uncertainty estimation, auNN is a competitive alternative in general practice.

### 3.3 COMPARISON WITH DGPS

As explained in Section 2, the choice of a GP prior for activation stochasticity establishes a strong connection with DGPs. The main difference is that auNN performs a linear projection from $D^{l-1}$ to $D^l$ dimensions before applying $D^l$ 1D GPs, whereas DGPs define $D^l$ GPs directly on the $D^{l-1}$ dimensional space. This means that auNN units are simpler than those of DGP, recall Figure 2a. Here we show two practical implications of this.

First, it is reasonable to hypothesise that DGP units may require a higher number of inducing points $M$ than auNN, since they need to cover a multi-dimensional input space. By contrast, auNN may require a higher number of hidden units $D$, since these are simpler. Importantly, the computational cost is not symmetric in $M$ and $D$, but significantly cheaper on $D$, recall Section 2. Figure 6b shows the performance of auNN and DGP for different values of $M$ and $D$ on the UCI Kin8 set (with one hidden layer; depth will be analyzed next). As expected, note the different influence by $M$ and $D$: whereas auNN improves "by rows" (i.e. as $D$ grows), DGP does it "by columns" (i.e. as $M$ grows)[5]. Next section (Section 3.4), will show that this makes auNN faster than DGP in practice. An analogous figure for RMSE and full numeric results are in Appendix C (Figure 9 and Tables 9-10).

Second, auNN simpler units might be better suited for deeper architectures. Figure 6c shows the performance on the UCI Power dataset when depth is additionally considered. It can be observed that auNN is able to take greater advantage of depth, which translates into better overall performance.

---

[5]Interestingly, the fact that DGP is not greatly influenced by $D$ could be appreciated in its recommended value in the original work (Salimbeni & Deisenroth, 2017). They set $D = \min(30, D^0)$, where $D^0$ is the input dimension. This limits $D$ to a maximum value of 30.

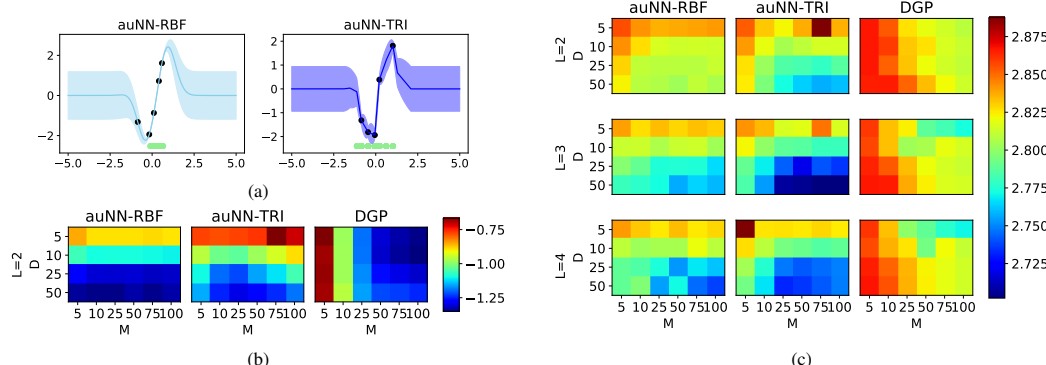

Figure 6: **(a)** One example of activation function (mean and standard deviation) learned by auNN with each kernel. RBF's one is smoother, whereas TRI's is piecewise linear, inspired by ReLu. Black dots represent (the mean of) the inducing point values. Green dots are the locations of input data when propagated to the corresponding unit. **(b)** Test NLL of auNN and DGP for different values of $M$ (number of inducing points) and $D$ (number of hidden units). The lower the better. The results are the average over five independent runs with different splits. Whereas DGP improves "by columns" (i.e. with $M$), auNN does it "by rows" (i.e. with $D$). This is as hypothesized, and is convenient from a scalability viewpoint. **(c)** Test NLL with increasing depth ($L = 2, 3, 4$). This supports that auNN might benefit more than DGP from deeper networks. Moreover, the aforementioned different influence of $M$ and $D$ on DGP and auNN is confirmed here.

Moreover, the aforementioned different influence of $D$ and $M$ on DGP and on auNN is also confirmed here. The results on RMSE are similar, see Figure 10 and Tables 11-12 in Appendix C.

Finally, it may be argued that auNN closely resembles a DGP with additive kernel (Duvenaud et al., 2011; Durrande et al., 2011) (DGP-add hereafter). Recall that an additive kernel models functions that are decomposed as $f(\mathbf{x}) = f_1(x_1) + \cdots + f_D(x_D)$. Therefore, the model for $a^{l+1}|a^l$ *in auNN* is very similar to that of $f^{l+1}|f^l$ *in DGP-add*, see Figure 11 in Appendix C. Specifically, in both cases, the input ($a^l$ in auNN, $f^l$ in DGP-add) goes through 1D GPs and then these are aggregated (linear combination through $\mathbf{W}$ in auNN, summation in DGP-add) to yield the output ($a^{l+1}$ in auNN, $f^{l+1}$ in DGP-add). However, there exists a key difference. In auNN, all the nodes in the $(l+1)$-th layer (i.e. $a_i^{l+1}$) aggregate a *shared* set of distinct functions (namely, $f_i^l$), each node using its own weights to aggregate them. While in DGP-add, there is not such shared set of functions, and each node in the $(l+1)$-th layer (i.e. $f_i^{l+1}$) aggregates a different set of GP realizations (i.e. the unlabelled blue nodes in Figure 11c). This subtle theoretical difference has empirical implications, since many more functions need to be learned for DGP-add. Indeed, Figures 12 and 13 in Appendix C compare the performance of DGP-add and auNN-RBF (the experimental setting is analogous to that of Figure 6c)[6]. We observe that the results obtained by DGP-add are worse than those by auNN-RBF, probably due to the larger number of functions that need to be learned in DGP-add.

### 3.4 CLASSIFICATION, SCALABILITY, AND ADDITIONAL METRICS

So far, we have experimented with small to medium regression datasets, and uncertainty estimation has been measured through the (negative) log likelihood and the visual inspection of the predictive distribution (Figures 3 and 5). Here we focus on two large scale classification datasets (up to $10^7$ instances), and additional metrics that account for uncertainty calibration are reported. We use the well-known particle physics binary classification sets HIGGS ($N = 11M$, $D = 28$) and SUSY ($N = 5M$, $D = 18$) (Baldi et al., 2014). We consider DGP as a baseline, as it obtained state-of-the-art results for these datasets (Salimbeni & Deisenroth, 2017). For all the methods, we consider a Robust-Max classification likelihood (Hernández-Lobato et al., 2011).

The metrics to be used are the Brier score (Gneiting & Raftery, 2007) and the Expected Calibration Error (ECE) (Guo et al., 2017). The former is a proper score function that measures the accuracy of probabilistic predictions for categorical variables. In practice, it is computed as the mean squared

---

[6]For a fair comparison, here we use auNN-RBF (and not TRI), because DGP-add leverages a RBF kernel.

Table 2: Brier score and expected calibration error (ECE) for auNN and DGP in the large scale classfication datasets HIGGS and SUSY (the lower the better in both metrics). The standard error (on three splits) is close to zero in all cases, see Table 13 in Appendix C.

| | | | | auNN | | | | | | DGP | | |
|---|---|---|---|---|---|---|---|---|---|---|---|---|
| | | N | D | RBF-2 | RBF-3 | RBF-4 | TRI-2 | TRI-3 | TRI-4 | DGP-2 | DGP-3 | DGP-4 |
| Brier | HIGGS | 11M | 28 | 0.3363 | 0.3159 | **0.3098** | 0.3369 | 0.3172 | 0.3118 | 0.4527 | 0.4399 | 0.4378 |
| | SUSY | 5M | 18 | 0.2746 | 0.2739 | **0.2737** | 0.2749 | 0.2742 | 0.2738 | 0.3815 | 0.3816 | 0.3804 |
| ECE | HIGGS | 11M | 28 | **0.2196** | 0.2383 | 0.2427 | 0.2198 | 0.2390 | 0.2397 | 0.4352 | 0.4303 | 0.4251 |
| | SUSY | 5M | 18 | **0.3453** | 0.3496 | 0.3504 | 0.3462 | 0.3485 | 0.3465 | 0.5304 | 0.5291 | 0.5273 |

Table 3: Average training time per batch over 50 independent runs (in seconds). The standard error is low in all cases, see Table 14 in Appendix C.

| | | auNN | | | | | | DGP | | |
|---|---|---|---|---|---|---|---|---|---|---|
| | RBF-2 | RBF-3 | RBF-4 | TRI-2 | TRI-3 | TRI-4 | DGP-2 | DGP-3 | DGP-4 |
| HIGGS | 0.0962 | 0.1607 | 0.2259 | **0.0922** | 0.1647 | 0.2308 | 0.1918 | 0.3102 | 0.3930 |
| SUSY | 0.0926 | 0.1564 | 0.2245 | **0.0923** | 0.1563 | 0.2265 | 0.1430 | 0.2129 | 0.2771 |

difference between a one dimensional vector with the probability for each class label and the one-hot encoding of the actual class. The latter measures miscalibration as the difference in expectation between confidence and accuracy. This is done by partitioning the predictions in $M$ equally spaced bins and taking a weighted average of the bins' accuracy/confidence difference, see (Guo et al., 2017, Eq.(3)) for details.

Table 2 shows the Brier score and ECE for auNN and DGP for different values of $L$ (depth). We observe that auNN outperforms DGP in both metrics, achieving superior uncertainty estimation. Both TRI and RBF kernels obtain similar results for auNN. Notice that the Brier score generally improves with the network depth, whereas the performance in ECE decreases with depth. Interestingly, this behavior was also observed for standard NNs (Guo et al., 2017, Figure 2a).

Finally, as was theoretically justified in Section 2, auNN can scale up to very large datasets (HIGGS has more than $10^7$ training instances). Regarding the practical computational cost, Table 3 shows the average training time per batch for both auNN and DGP in the previous datasets. Although the theoretical complexity is analogous for both methods (recall Section 2), the experiments in Figures 6b-c showed that DGP requires larger values of $M$, whereas auNN needs larger $D$ [7]. Since the computational cost is not symmetric on $M$ and $D$, but significantly cheaper in the latter (recall Section 2), auNN is faster than DGP in practice.

## 4  RELATED WORK

Activation-level uncertainty is introduced here as an alternative to weight-space stochasticity. The expressiveness of the latter has been recently analyzed in the recent work (Wenzel et al., 2020), where the authors advocate a modified BNN objective. Alternatively, different prior specifications are studied in (Hafner et al., 2020; Pearce et al., 2019; Flam-Shepherd et al., 2017), in addition to the fBNN discussed here (Sun et al., 2019). However, none of these works consider stochasticity on the activations.

Since we present a straightforward use of VI for auNN, in this work we have compared empirically with the well-known VI-based BBP for BNNs. Yet, we expect auNN to benefit from independent inference refinements like those proposed over the last years for BNNs. For instance, natural-gradient VI allows for leveraging techniques such as BatchNorm or data augmentation (Osawa et al., 2019), and the information contained in the SGD trajectory can be exploited as well (Maddox et al., 2019). Also, getting rid of the gradient variance through deterministic approximate moments has provided enhanced results in BNNs (Wu et al., 2019).

---

[7]In this section, both DGP and auNN are trained with one hidden layer and their optimal configuration according to the previous experiment: large $M$ for DGP ($M = 100$, $D$ is set as recommended by the authors, i.e. $D = \min(30, D^0)$), and large $D$ for auNN ($D = 50$, $M$ is set to the intermediate value of $M = 25$).

A key aspect of auNN is the modelling of the activation function. This element of neural nets has been analyzed before. For instance, self-normalizing neural nets (Klambauer et al., 2017) induce the normalization that is explicitly performed in related approaches such as BatchNorm (Ioffe & Szegedy, 2015) and weight and layer normalization (Salimans & Kingma, 2016; Ba et al., 2016). Learnable deterministic activations have been explored too, e.g. (He et al., 2015; Agostinelli et al., 2014). However, as opposed to auNN, in all these cases the activations are deterministic.

Probabilistic neural networks such as Natural-Parameter Networks (NPN) (Wang et al., 2016) propagate probability distributions through layers of transformations. Therefore, the values of the activations are also described by probability distributions (specifically, the exponential family is used in NPN). Fast dropout training (Wang & Manning, 2013) and certain variants of NPNs can be also viewed in this way (Shekhovtsov & Flach, 2018; Postels et al., 2019). However, in auNN the activations are modelled themselves as stochastic learnable components that follow a GP prior. Along with the deterministic weights, this provides a conceptually different approach to model uncertainty.

A very preliminary study on GP-based activation functions is proposed in (Urban & van der Smagt, 2018). However, the method is not empirically evaluated, no connection with deep GPs is provided, and the inference approach is limited. Namely, the output of each unit is approximated with a Gaussian whose mean and covariance are computed in closed-form, as was done in (Bui et al., 2016) for DGPs. However, this is only tractable for the RBF kernel (in particular, it cannot leverage the more convenient TRI kernel studied here), and the Gaussian approximation typically yields worse results than Monte Carlo approximations to the ELBO as used here (indeed, DSVI (Salimbeni & Deisenroth, 2017) substantially improved the results for DGPs compared to (Bui et al., 2016)).

## 5 Conclusions and Future Work

We proposed a novel approach for uncertainty estimation in neural network architectures. Whereas previous methods are mostly based on a Bayesian treatment of the weights, here we move the stochasticity to the activation functions, which are modelled with a simple 1D GP and a triangular kernel inspired by the ReLu. Our experiments show that the proposed method obtains better calibrated uncertainty estimates and is competitive in standard prediction tasks. Moreover, the connection with deep GPs is analyzed. Namely, our approach requires fewer inducing points and is better suited for deep architectures, achieving superior performance.

We hope this work raises interest on alternative approaches to model uncertainty in neural networks. One of the main directions of future research is to deeply understand the properties induced by each one of the kernels considered here (i.e. the triangular one and RBF). In particular, it would be interesting to automatically learn the optimal kernel for each unit in a probabilistic way. Also, the use of a GP prior for the activation function may hamper the scalability of auNN to wider and/or deeper networks. In these cases, the GP-based activation model could be substituted by a simpler Bayesian parametric one. This would allow for a cheaper modelling of uncertainty within the activations. Finally, since only the activation function is modified, important deep learning elements such as convolutional layers can be still incorporated.

## Acknowledgements

This work was supported by the "Agencia Estatal de Investigación" of the Spanish "Ministerio de Ciencia e Innovación" under contract PID2019-105142RB-C22/AEI/10.13039/501100011033, and the Spanish "Ministerio de Economía, Industria y Competitividad" under contract DPI2016-77869-C2-2-R. DHL acknowledges support from the Spanish "Ministerio de Ciencia e Innovación" (projects TIN2016-76406-P and PID2019-106827GB-I00/AEI/10.13039/501100011033). PMA was funded by La Caixa Banking Foundation (ID 100010434, Barcelona, Spain) through La Caixa Fellowship for Doctoral Studies LCF/BQ/ES17/11600011, and the University of Granada through the program "Proyectos de Investigación Precompetitivos para Jóvenes Investigadores del Plan Propio 2019" (ref. PPJIB2019-03).

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

## A  PRACTICAL SPECIFICATIONS FOR AUNN

**Whitening transformation for** $q(\mathbf{u}_d^l)$**.** The proposed parametric posterior for each unit is given by the Gaussian $q(\mathbf{u}_d^l) = \mathcal{N}(\mathbf{u}_d^l | \mathbf{m}_d^l, \mathbf{S}_d^l)$. The GP prior on $\mathbf{u}_d^l$ is $p(\mathbf{u}_d^l) = \mathcal{N}(\mathbf{u}_d^l | \boldsymbol{\mu}_d^l, \mathbf{K}_d^l)$, with $\boldsymbol{\mu}_d^l = \mu_d^l(\mathbf{z}_d^l)$ and $\mathbf{K}_d^l = k_d^l(\mathbf{z}_d^l, \mathbf{z}_d^l)$. For numerical stability and to reduce the amount of operations, we use a white representation for $q(\mathbf{u}_d^l)$, as is common practice in (D)GPs (De G. Matthews et al., 2017; Salimbeni & Deisenroth, 2017). That is, we consider the variable $\mathbf{v}_d^l \sim \mathcal{N}(\tilde{\mathbf{m}}_d^l, \tilde{\mathbf{S}}_d^l)$, with $\mathbf{u}_d^l = \boldsymbol{\mu}_d^l + (\mathbf{K}_d^l)^{1/2} \mathbf{v}_d^l$. Specifically, in the code the variable $\tilde{\mathbf{m}}_d^l$ is denoted as `q_mu`, and $\tilde{\mathbf{S}}_d^l$ is represented through its Cholesky factorization $(\tilde{\mathbf{S}}_d^l)^{1/2}$, which is named `q_sqrt`.

**Initialization of the variational parameters** $\{\mathbf{m}_d^l\}$**.** These are the mean of the posterior distribution on the inducing points. Therefore, their value determines the *initialization of the activation function*. If the RBF kernel is used, $\{\mathbf{m}_d^l\}$ are initialized to the prior $\boldsymbol{\mu}_d^l = \mu_d^l(\mathbf{z}_d^l)$ (since we are using the aforementioned white representation, `q_mu` is initialized to zero). This is the most standard initialization in GP literature. For the TRI kernel, $\{\mathbf{m}_d^l\}$ are initialized according to the ReLu which TRI is inspired by, i.e. $\mathbf{m}_d^l = \text{ReLu}(\mathbf{z}_d^l)$.

**Initialization of the variational parameters** $\{\mathbf{S}_d^l\}$**.** The posterior distribution covariance matrices are initialized to the prior $\mathbf{K}_d^l = k_d^l(\mathbf{z}_d^l, \mathbf{z}_d^l)$ (that is, `q_sqrt` is initialized to the identity matrix). Following common practise for DGPs (Salimbeni & Deisenroth, 2017), the covariance matrices of inner layers are scaled by $10^{-5}$.

**Initialization of the weights.** The Glorot uniform initializer (Glorot & Bengio, 2010), also called Xavier uniform initializer, is used for the weights. The biases are initialized to zero.

**Initialization of the kernel hyperparameters.** The kernels used (RBF and TRI) have two hyperparameters: the variance $\gamma$ and the lengthscale $\ell$. Both are always initialized to 1 (except for the lengthscale in the 1D example in Section 3.1, where $\ell$ is initialized to 0.1).

**Initialization of the inducing points.** In order to initialize $\mathbf{z}_d^l$, the $N$ input data points are propagated through the network with the aforementioned initial weights, biases, and activation function. Then, in each layer and unit, $\mathbf{z}_d^l$ is initialized with a `linspace` between the minimum and maximum of the $N$ values there (the minimum (resp. the maximum) is decreased (resp. increased) by 0.1 to strictly contain the interval of interest).

**Initialization of the regression likelihood noise**. In the regression problems, we use a Gaussian likelihood $p(y|f) = \mathcal{N}(y|f, \sigma^2)$. The standard deviation of the noise is initialized to $\sigma = 0.1$.

**Mean function.** We always use a zero mean function. Since data is normalized to have zero mean (and standard deviation equal to one), a zero mean function allows for reverting to the empirical mean for OOD data, as explained in the main text.

**Optimizer and learning rate.** Throughout the work, we use the Adam Optimizer (Kingma & Ba, 2014) with default parameters and learning rate of 0.001.

## B  EXPERIMENTAL DETAILS FOR THE EXPERIMENTS

All the experiments were run on a NVIDIA Tesla P100. In order to predict, all the methods utilize 100 test samples in all the experiments. Details for each section are provided below.

**An illustrative example (Section 3.1 in the main text).** All the methods use two layers (i.e. one hidden layer). The hidden layer has $D = 25$ units in all cases. BNN and fBNN use ReLu activations. The auNN methods use $M = 10$ inducing points in each unit (the rest of methods do not have such inducing points). The methods are trained during 5000 epochs with the whole dataset (no mini-batches). The dataset is synthetically generated to have two clusters of points around $x = \pm 1$. More specifically, 30 points are sampled uniformly in each interval $(x - 0.3, x + 0.3)$ for $x = \pm 1$, and the output is given by the sin function plus a Gaussian noise of standard deviation 0.1. We have also trained DGP and GP on this dataset, see Figure 14. Both methods use $M = 10$ inducing points, and are trained during 5000 epochs with the whole dataset (no mini-batches). DGP has one one hidden layer with $D = 25$ units.

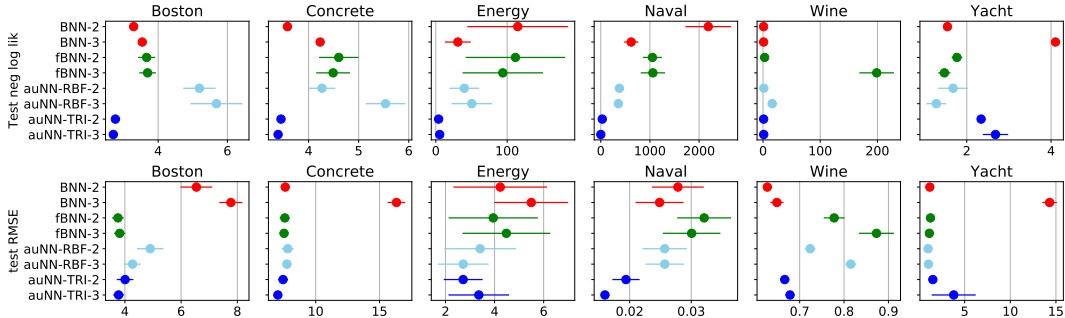

Figure 7: Performance of the compared methods in the gap splits for six UCI datasets. Mean and one standard error of NLL (upper row) and RMSE (lower row) are shown, the lower the better.

**UCI regression datasets with gap (and standard) splits (Section 3.2 in the main text).** The methods use $L = 2, 3$ layers. In all cases, the hidden layers have $D = 50$ units. BNN and fBNN use ReLu activations. The methods are trained during 10000 epochs, with a mini-batch size that depends on the size of the dataset. For those with fewer than 5000 instances (i.e. Boston, Concrete, Energy, Wine and Yacht), the mini-batch size is 500. For those with more than 5000 (i.e. Naval), the mini-batch size is 5000. Recall from the main text that each dataset has as many gap splits as dimensionality, with 2/3 for train and 1/3 for test. In the case of standard splits, each dataset uses 10 random 90%-10% train-test splits. Regarding the segment used in Figure 5, each extreme of the segment is a point from a different connected component of the training set. These are chosen so that the function is well-known in the extremes (but not along the segment, which crosses the gap). Namely, the extremes are chosen as the training points who have minimum average distance to the closest five points in its connected component.

**Comparison with DGPs (Section 3.3 in the main text).** Here, different values of depth $L$, number of inducing points $M$ and number of hidden layers $D$ are studied (see the main text). auNN is trained during 5000 epochs, with a mini-batch size of 5000 (20000 epochs are used for DGP, as proposed by the authors (Salimbeni & Deisenroth, 2017)). Each experiment is repeated on five random 90%-10% train-test splits. DGP uses a RBF kernel. The experimental details for DGP-add are the same as for DGP, with the only difference of the kernel. Namely, an additive kernel using RBF components is used for DGP-add.

**Large scale experiments (Section 3.4 in the main text).** Since we are dealing with classification datasets, a Robust-Max likelihood is used in all cases (Hernández-Lobato et al., 2011). The values of $D$ and $M$ are chosen following the conclusions from Section 3.3. That is, DGP needs large $M$ (the largest $M = 100$ is used), but is less influenced by $D$ (this is chosen as recommended by the authors (Salimbeni & Deisenroth, 2017): $D = \min(30, D^0)$, with $D^0$ the dimensionality of the input data). auNN needs large $D$ (the largest $D = 50$ is used), but is less influenced by $M$ (the intermediate value $M = 25$ is chosen). All the methods are trained during 100 epochs, with a mini-batch size of 5000. Three random train-test splits are used. In both datasets, 500000 instances are used for test (which leaves 10.5M and 4.5M training instances for HIGGS and SUSY, respectively).

## C ADDITIONAL FIGURES AND TABLES

Finally, additional material is provided here. Every figure and table is referred from the main text.

Table 4: Test NLL for the gap splits of the six UCI datasets (mean and one standard error, the lower the better). Last column is the per-group (weight-space stochasticity vs activation-level stochasticity) average rank.

|  | Boston | Concrete | Energy | Naval | Wine | Yacht | Rank | Rank (group) |
|---|---|---|---|---|---|---|---|---|
| BNN-2 | 3.29±0.10 | 3.58±0.09 | 114.84±70.69 | 2186.30±464.32 | **0.96±0.01** | 1.54±0.09 | 3.92±0.79 | |
| BNN-3 | 3.54±0.03 | 4.23±0.04 | 30.91±19.97 | 618.44±147.99 | 0.98±0.02 | 4.10±0.03 | 4.98±0.70 | 4.83±0.32 |
| fBNN-2 | 3.67±0.25 | 4.60±0.39 | 111.65±69.68 | 1050.65±192.61 | 2.80±0.31 | 1.77±0.12 | 5.04±0.36 | |
| fBNN-3 | 3.69±0.24 | 4.49±0.34 | 93.92±56.45 | 1060.54±247.21 | 198.76±30.24 | 1.47±0.15 | 5.36±0.50 | |
| auNN-RBF-2 | 5.19±0.47 | 4.27±0.26 | 39.93±20.89 | 379.55±67.74 | 1.44±0.05 | 1.68±0.35 | 4.69±0.61 | |
| auNN-RBF-3 | 5.68±0.75 | 5.54±0.40 | 50.48±28.26 | 352.94±72.13 | 16.05±1.13 | **1.28±0.23** | 5.29±0.89 | **4.17±0.40** |
| auNN-TRI-2 | 2.77±0.06 | 3.45±0.06 | **3.99±1.14** | 30.47±5.54 | 1.06±0.03 | 2.34±0.03 | **3.25±0.57** | |
| auNN-TRI-3 | **2.70±0.04** | **3.39±0.06** | 5.50±2.45 | **2.38±3.23** | 1.23±0.04 | 2.68±0.30 | 3.47±0.80 | |

Table 5: Test RMSE for the gap splits of the six UCI datasets (mean and one standard error, the lower the better). Last column is the per-group (weight-space stochasticity vs activation-level stochasticity) average rank.

|  | Boston | Concrete | Energy | Naval | Wine | Yacht | Rank | Rank (group) |
|---|---|---|---|---|---|---|---|---|
| BNN-2 | 6.54±0.56 | 7.62±0.35 | 4.23±1.91 | 0.03±0.00 | **0.63±0.01** | 1.18±0.11 | 4.09±0.67 | |
| BNN-3 | 7.77±0.40 | 16.33±0.67 | 5.27±1.41 | 0.02±0.00 | 0.64±0.01 | 14.31±0.76 | 6.15±0.91 | 4.91±0.37 |
| fBNN-2 | **3.75±0.21** | 7.58±0.41 | 3.95±1.82 | 0.03±0.00 | 0.78±0.02 | 1.25±0.08 | 4.70±0.54 | |
| fBNN-3 | 3.81±0.20 | 7.52±0.36 | 4.48±1.79 | 0.03±0.00 | 0.87±0.04 | 1.13±0.12 | 4.71±0.53 | |
| auNN-RBF-2 | 4.90±0.47 | 7.81±0.47 | 3.41±1.46 | 0.03±0.00 | 0.72±0.01 | **0.99±0.18** | 4.32±0.47 | |
| auNN-RBF-3 | 4.27±0.29 | 7.74±0.21 | 2.72±1.03 | 0.03±0.00 | 0.82±0.01 | 1.03±0.14 | 4.27±0.55 | **4.09±0.23** |
| auNN-TRI-2 | 4.01±0.30 | 7.44±0.38 | **2.72±0.79** | 0.02±0.00 | 0.67±0.01 | 1.51±0.20 | 3.90±0.33 | |
| auNN-TRI-3 | 3.78±0.19 | **7.03±0.23** | 3.36±1.23 | **0.02±0.00** | 0.68±0.01 | 3.80±2.41 | **3.85±0.46** | |

Table 6: Test NLL for the standard splits of the six UCI datasets (mean and one standard error, the lower the better). Last column is the per-group (weight-space stochasticity vs activation-level stochasticity) average rank.

| test NLL | Boston | Concrete | Energy | Naval | Wine | Yacht | Rank | Rank (group) |
|---|---|---|---|---|---|---|---|---|
| BNN-2 | 2.71±0.07 | 3.12±0.02 | 0.65±0.04 | -5.38±0.59 | 0.99±0.02 | 1.01±0.07 | **3.78±0.41** | |
| BNN-3 | 3.62±0.05 | 4.24±0.01 | 0.80±0.03 | -5.02±0.33 | 1.01±0.02 | 4.06±0.05 | 6.25±0.70 | 4.5±0.39 |
| fBNN-2 | 2.83±0.20 | 3.20±0.04 | 0.67±0.04 | -6.17±0.02 | 1.55±0.08 | 0.77±0.02 | 4.13±0.57 | |
| fBNN-3 | 2.75±0.14 | 3.13±0.05 | 0.65±0.03 | **-6.26±0.00** | 207.43±9.12 | 0.79±0.02 | 3.83±0.85 | |
| auNN-RBF-2 | 3.38±0.30 | 3.14±0.05 | 0.63±0.03 | -5.40±0.08 | 1.16±0.06 | **0.52±0.04** | 3.97±0.60 | |
| auNN-RBF-3 | 3.89±0.47 | 3.25±0.13 | **0.53±0.07** | -5.69±0.03 | 8.98±1.51 | 0.54±0.03 | 4.42±0.85 | 4.5±0.43 |
| auNN-TRI-2 | 2.56±0.05 | 3.08±0.02 | 1.47±0.04 | -4.81±0.07 | **0.96±0.03** | 2.25±0.02 | 4.78±0.92 | |
| auNN-TRI-3 | **2.50±0.02** | **2.98±0.02** | 1.42±0.02 | -3.43±0.32 | 1.10±0.07 | 2.26±0.01 | 4.83±1.01 | |

Table 7: Test RMSE for the standard splits of the six UCI datasets (mean and one standard error, the lower the better). Last column is the per-group (weight-space stochasticity vs activation-level stochasticity) average rank.

| test RMSE | Boston | Concrete | Energy | Naval | Wine | Yacht | Rank | Rank (group) |
|---|---|---|---|---|---|---|---|---|
| BNN-2 | 3.47±0.34 | 5.49±0.13 | 0.45±0.02 | 0.00±0.00 | 0.65±0.01 | 0.68±0.08 | 4.70±0.48 | |
| BNN-3 | 8.89±0.45 | 16.71±0.20 | 0.51±0.02 | 0.00±0.00 | 0.67±0.02 | 13.49±0.94 | 6.50±0.64 | 4.59±0.41 |
| fBNN-2 | 2.80±0.21 | 5.34±0.13 | 0.47±0.02 | 0.00±0.00 | 0.70±0.02 | **0.33±0.04** | 3.70±0.61 | |
| fBNN-3 | **2.74±0.16** | 5.07±0.12 | 0.46±0.02 | **0.00±0.00** | 0.83±0.02 | 0.36±0.04 | 3.45±0.88 | |
| auNN-RBF-2 | 3.16±0.23 | 5.13±0.16 | 0.45±0.02 | 0.00±0.00 | 0.67±0.02 | 0.41±0.04 | 4.25±0.35 | |
| auNN-RBF-3 | 3.01±0.25 | **4.51±0.18** | **0.41±0.03** | 0.00±0.00 | 0.76±0.02 | 0.38±0.03 | **3.35±0.77** | 4.41±0.41 |
| auNN-TRI-2 | 3.00±0.26 | 5.21±0.10 | 0.72±0.02 | 0.00±0.00 | **0.62±0.02** | 1.15±0.14 | 5.40±0.80 | |
| auNN-TRI-3 | 2.81±0.17 | 4.67±0.15 | 0.65±0.03 | 0.01±0.00 | 0.62±0.02 | 1.16±0.15 | 4.65±1.00 | |

Table 8: Standard error obtained by auNN and DGP in three splits of the large scale classification datasets HIGGS and SUSY.

|  |  |  | auNN | | | | | | DGP | | |
|---|---|---|---|---|---|---|---|---|---|---|---|
|  | N | D | RBF-2 | RBF-3 | RBF-4 | TRI-2 | TRI-3 | TRI-4 | DGP-2 | DGP-3 | DGP-4 |
| HIGGS | 11M | 28 | 0.0001 | 0.0006 | 0.0007 | 0.0003 | 0.0004 | 0.0008 | 0.0005 | 0.0009 | 0.0010 |
| SUSY | 5M | 18 | 0.0004 | 0.0005 | 0.0005 | 0.0005 | 0.0005 | 0.0004 | 0.0005 | 0.0027 | 0.0035 |

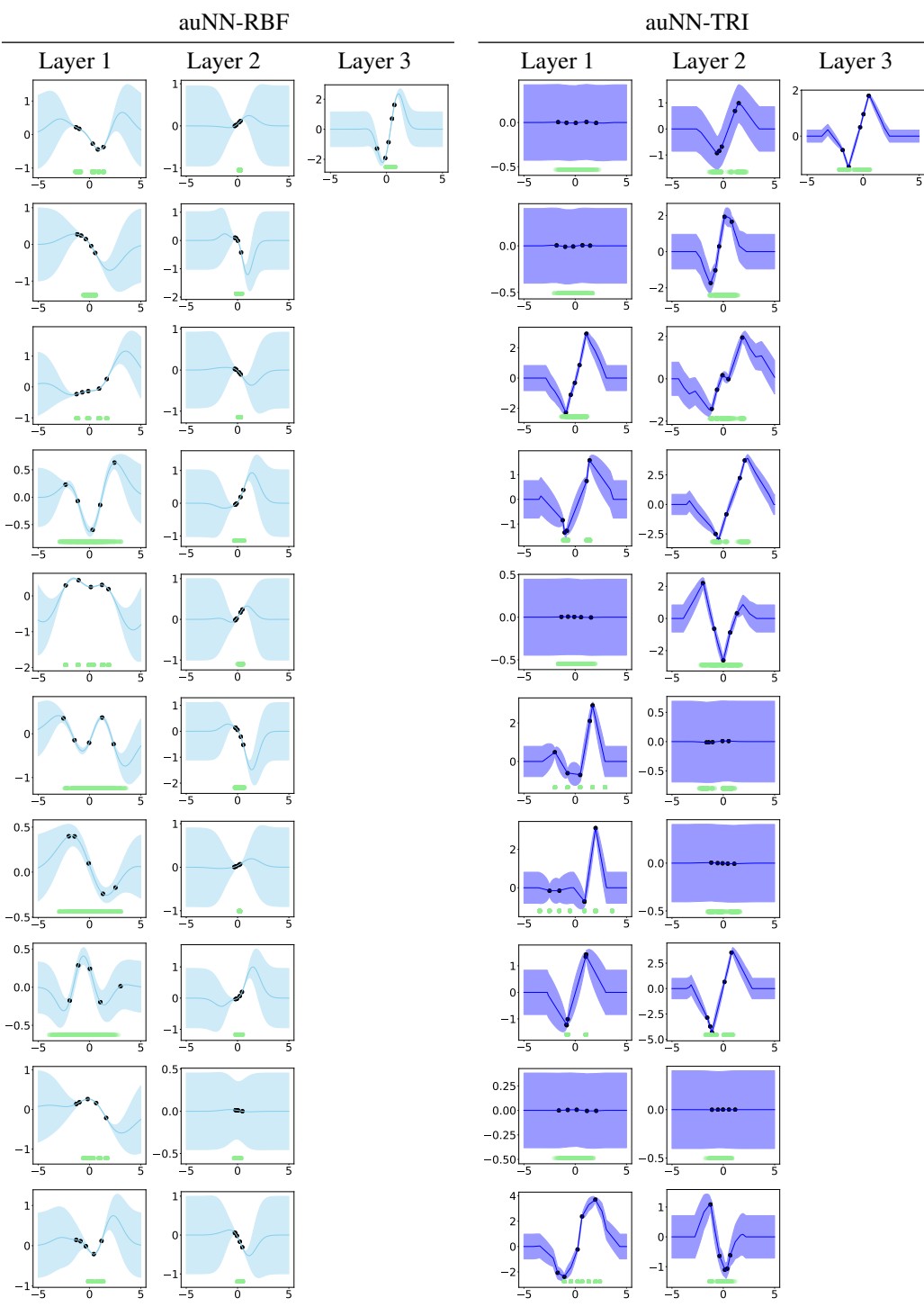

Figure 8: A complete example of the activation functions learned by auNN with RBF and TRI kernels. These were obtained for the Energy dataset with the first gap split, using three layers, 10 hidden units per (hidden) layer, and 5 inducing points in each unit. Whereas auNN-RBF learns smoother activations, auNN-TRI ones are piece-wise linear, inspired by the ReLu. Notice that auNN allows units to switch off if they are not required. Black dots represent the five inducing points in each unit. Green points are the locations of the input data when propagated to the corresponding unit.

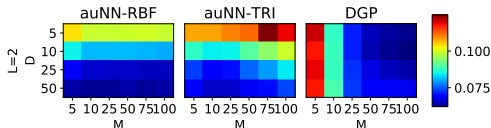

Figure 9: Test RMSE of auNN and DGP for different values of $M$ (number of inducing points) and $D$ (number of hidden units). Results are the average over 5 independent runs on the UCI Kin8 dataset. The lower the better. Whereas DGP improves "by columns" (i.e. with $M$), auNN does it "by rows" (i.e. with $D$). This is as theoretically expected, and it is convenient from a scalability viewpoint.

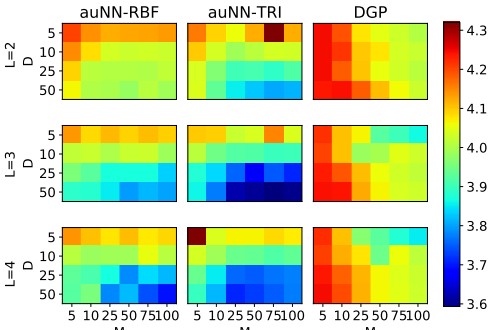

Figure 10: Test RMSE with increasing depth ($L = 2, 3, 4$). This supports that auNN might benefit more than DGP from deeper networks. Moreover, the aforementioned different influence of $M$ and $D$ on DGP and auNN is confirmed here.

Table 9: Test NLL of auNN and DGP for different values of $M$ (number of inducing points) and $D$ (number of hidden units). Mean and one standard error over 5 independent runs on the UCI Kin8 dataset are shown. The lower the better.

| | | auNN-RBF | | | | | | auNN-TRI | | | | | | DGP | | | | | |
|---|---|---|---|---|---|---|---|---|---|---|---|---|---|---|---|---|---|---|---|
| D | M | 5 | 10 | 25 | 50 | 75 | 100 | 5 | 10 | 25 | 50 | 75 | 100 | 5 | 10 | 25 | 50 | 75 | 100 |
| 5 | | -0.85±0.01 | -0.89±0.01 | -0.89±0.01 | -0.89±0.01 | -0.89±0.01 | -0.90±0.01 | -0.78±0.00 | -0.79±0.03 | -0.78±0.05 | -0.77±0.04 | -0.67±0.06 | -0.71±0.04 | -0.67±0.01 | -0.98±0.00 | -1.19±0.01 | -1.30±0.01 | -1.33±0.01 | -1.34±0.01 |
| 10 | | -1.06±0.01 | -1.09±0.01 | -1.09±0.01 | -1.09±0.01 | -1.10±0.02 | -1.10±0.01 | -0.96±0.01 | -1.02±0.01 | -1.03±0.01 | -0.98±0.03 | -0.94±0.03 | -0.89±0.03 | -0.69±0.01 | -0.98±0.00 | -1.19±0.00 | -1.30±0.01 | -1.33±0.01 | -1.35±0.01 |
| 25 | | -1.27±0.02 | -1.30±0.02 | -1.30±0.02 | -1.30±0.02 | -1.31±0.01 | -1.31±0.02 | -1.09±0.01 | -1.19±0.01 | -1.22±0.01 | -1.15±0.02 | -1.11±0.01 | -1.06±0.03 | -0.68±0.01 | -0.98±0.00 | -1.17±0.01 | -1.26±0.01 | -1.29±0.01 | -1.30±0.01 |
| 50 | | -1.33±0.01 | -1.34±0.01 | -1.34±0.02 | -1.33±0.01 | -1.34±0.02 | -1.32±0.03 | -1.15±0.01 | -1.24±0.01 | -1.29±0.01 | -1.26±0.01 | -1.24±0.02 | -1.19±0.02 | -0.69±0.01 | -0.96±0.01 | -1.16±0.01 | -1.21±0.01 | -1.22±0.01 | -1.24±0.01 |

Table 10: Test RMSE of auNN and DGP for different values of $M$ (number of inducing points) and $D$ (number of hidden units). Mean and one standard error over 5 independent runs on the UCI Kin8 dataset are shown. The lower the better.

| | | auNN-RBF | | | | | | auNN-TRI | | | | | | DGP | | | | | |
|---|---|---|---|---|---|---|---|---|---|---|---|---|---|---|---|---|---|---|---|
| D | M | 5 | 10 | 25 | 50 | 75 | 100 | 5 | 10 | 25 | 50 | 75 | 100 | 5 | 10 | 25 | 50 | 75 | 100 |
| 5 | | 0.10±0.00 | 0.10±0.00 | 0.10±0.00 | 0.10±0.00 | 0.10±0.00 | 0.10±0.00 | 0.11±0.00 | 0.11±0.00 | 0.11±0.01 | 0.11±0.00 | 0.12±0.01 | 0.12±0.00 | 0.12±0.00 | 0.09±0.00 | 0.07±0.00 | 0.07±0.00 | 0.06±0.00 | 0.06±0.00 |
| 10 | | 0.08±0.00 | 0.08±0.00 | 0.08±0.00 | 0.08±0.00 | 0.08±0.00 | 0.08±0.00 | 0.09±0.00 | 0.08±0.00 | 0.08±0.00 | 0.09±0.00 | 0.09±0.00 | 0.10±0.00 | 0.12±0.00 | 0.09±0.00 | 0.07±0.00 | 0.06±0.00 | 0.06±0.00 | 0.06±0.00 |
| 25 | | 0.07±0.00 | 0.07±0.00 | 0.07±0.00 | 0.07±0.00 | 0.07±0.00 | 0.07±0.00 | 0.07±0.00 | 0.07±0.00 | 0.07±0.00 | 0.08±0.00 | 0.08±0.00 | 0.08±0.00 | 0.12±0.00 | 0.09±0.00 | 0.07±0.00 | 0.07±0.00 | 0.07±0.00 | 0.06±0.00 |
| 50 | | 0.06±0.00 | 0.06±0.00 | 0.06±0.00 | 0.06±0.00 | 0.06±0.00 | 0.06±0.00 | 0.07±0.00 | 0.07±0.00 | 0.07±0.00 | 0.07±0.00 | 0.07±0.00 | 0.07±0.00 | 0.12±0.00 | 0.09±0.00 | 0.07±0.00 | 0.07±0.00 | 0.07±0.00 | 0.07±0.00 |

Table 11: Test NLL of auNN and DGP for different values of $M$ (number of inducing points) and $D$ (number of hidden units) as the depth increases from $L = 2$ to $L = 4$. Mean and one standard error over 5 independent runs on the UCI Power dataset are shown. The lower the better.

| | | | auNN-RBF | | | | | | auNN-TRI | | | | | | DGP | | | | | |
|---|---|---|---|---|---|---|---|---|---|---|---|---|---|---|---|---|---|---|---|---|
| L | D | M | 5 | 10 | 25 | 50 | 75 | 100 | 5 | 10 | 25 | 50 | 75 | 100 | 5 | 10 | 25 | 50 | 75 | 100 |
| 2 | 5 | | 2.86±0.02 | 2.84±0.02 | 2.84±0.02 | 2.84±0.02 | 2.84±0.02 | 2.84±0.02 | 2.85±0.02 | 2.83±0.02 | 2.82±0.02 | 2.84±0.02 | 2.89±0.03 | 2.84±0.02 | 2.87±0.02 | 2.85±0.02 | 2.83±0.02 | 2.82±0.02 | 2.81±0.02 | 2.81±0.02 |
| | 10 | | 2.84±0.02 | 2.83±0.02 | 2.82±0.02 | 2.81±0.02 | 2.81±0.02 | 2.81±0.02 | 2.84±0.02 | 2.82±0.02 | 2.80±0.02 | 2.81±0.02 | 2.81±0.02 | 2.81±0.02 | 2.87±0.02 | 2.86±0.02 | 2.83±0.02 | 2.83±0.02 | 2.82±0.02 | 2.81±0.02 |
| | 25 | | 2.83±0.02 | 2.81±0.02 | 2.81±0.02 | 2.81±0.02 | 2.81±0.02 | 2.81±0.02 | 2.83±0.02 | 2.80±0.02 | 2.78±0.02 | 2.78±0.02 | 2.78±0.02 | 2.79±0.02 | 2.87±0.02 | 2.85±0.02 | 2.83±0.02 | 2.82±0.02 | 2.81±0.02 | 2.81±0.02 |
| | 50 | | 2.82±0.02 | 2.81±0.02 | 2.81±0.02 | 2.80±0.02 | 2.81±0.02 | 2.81±0.02 | 2.82±0.02 | 2.80±0.02 | 2.77±0.02 | 2.76±0.02 | 2.76±0.02 | 2.76±0.03 | 2.86±0.02 | 2.87±0.02 | 2.85±0.02 | 2.83±0.02 | 2.82±0.02 | 2.81±0.02 |
| 3 | 5 | | 2.84±0.02 | 2.83±0.02 | 2.83±0.02 | 2.83±0.03 | 2.83±0.02 | 2.83±0.02 | 2.84±0.02 | 2.83±0.02 | 2.82±0.02 | 2.82±0.02 | 2.85±0.02 | 2.82±0.02 | 2.86±0.02 | 2.83±0.02 | 2.82±0.02 | 2.79±0.02 | 2.78±0.02 | 2.77±0.01 |
| | 10 | | 2.81±0.02 | 2.81±0.02 | 2.81±0.02 | 2.81±0.02 | 2.81±0.02 | 2.80±0.02 | 2.82±0.02 | 2.80±0.02 | 2.79±0.02 | 2.79±0.02 | 2.78±0.02 | 2.78±0.02 | 2.86±0.02 | 2.83±0.02 | 2.80±0.02 | 2.81±0.02 | 2.82±0.02 | 2.81±0.02 |
| | 25 | | 2.80±0.02 | 2.79±0.02 | 2.77±0.02 | 2.77±0.02 | 2.77±0.02 | 2.77±0.02 | 2.79±0.02 | 2.77±0.02 | 2.74±0.02 | 2.72±0.02 | 2.74±0.03 | 2.74±0.03 | 2.86±0.02 | 2.85±0.02 | 2.83±0.02 | 2.82±0.02 | 2.81±0.02 | 2.81±0.02 |
| | 50 | | 2.78±0.02 | 2.78±0.02 | 2.77±0.02 | 2.76±0.02 | 2.76±0.02 | 2.76±0.03 | 2.78±0.02 | 2.75±0.02 | 2.71±0.02 | 2.71±0.03 | 2.70±0.03 | 2.70±0.02 | 2.87±0.02 | 2.87±0.02 | 2.84±0.02 | 2.82±0.02 | 2.82±0.02 | 2.81±0.02 |
| 4 | 5 | | 2.84±0.02 | 2.83±0.02 | 2.82±0.02 | 2.83±0.02 | 2.82±0.01 | 2.83±0.02 | 3.69±0.35 | 2.83±0.01 | 2.83±0.02 | 2.83±0.02 | 2.83±0.02 | 2.82±0.02 | 2.86±0.02 | 2.83±0.02 | 2.80±0.02 | 2.79±0.02 | 2.78±0.02 | 2.77±0.02 |
| | 10 | | 2.81±0.02 | 2.80±0.02 | 2.80±0.02 | 2.82±0.02 | 2.82±0.02 | 2.82±0.02 | 2.83±0.02 | 2.81±0.01 | 2.79±0.01 | 2.79±0.02 | 2.79±0.02 | 2.79±0.02 | 2.86±0.02 | 2.84±0.02 | 2.83±0.02 | 2.79±0.02 | 2.82±0.02 | 2.81±0.02 |
| | 25 | | 2.79±0.02 | 2.78±0.02 | 2.77±0.02 | 2.75±0.02 | 2.77±0.02 | 2.76±0.02 | 2.80±0.01 | 2.78±0.02 | 2.75±0.02 | 2.74±0.02 | 2.75±0.03 | 2.75±0.02 | 2.86±0.02 | 2.85±0.02 | 2.83±0.02 | 2.82±0.02 | 2.82±0.02 | 2.81±0.02 |
| | 50 | | 2.79±0.02 | 2.80±0.02 | 2.75±0.03 | 2.77±0.03 | 2.75±0.03 | 2.74±0.03 | 2.79±0.01 | 2.77±0.01 | 2.73±0.02 | 2.74±0.02 | 2.74±0.02 | 2.75±0.02 | 2.87±0.02 | 2.85±0.02 | 2.84±0.02 | 2.82±0.02 | 2.82±0.02 | 2.81±0.02 |

Table 12: Test RMSE of auNN and DGP for different values of $M$ (number of inducing points) and $D$ (number of hidden units) as the depth increases from $L = 2$ to $L = 4$. Mean and one standard error over 5 independent runs on the UCI Power dataset are shown. The lower the better.

| L | D | M | auNN-RBF | | | | | | auNN-TRI | | | | | | DGP | | | | | |
|---|---|---|---|---|---|---|---|---|---|---|---|---|---|---|---|---|---|---|---|---|
| | | | 5 | 10 | 25 | 50 | 75 | 100 | 5 | 10 | 25 | 50 | 75 | 100 | 5 | 10 | 25 | 50 | 75 | 100 |
| 2 | 5 | | 4.20±0.09 | 4.14±0.08 | 4.12±0.08 | 4.13±0.08 | 4.13±0.07 | 4.14±0.09 | 4.16±0.08 | 4.09±0.08 | 4.06±0.09 | 4.12±0.09 | 4.32±0.13 | 4.12±0.09 | 4.24±0.10 | 4.19±0.09 | 4.08±0.09 | 4.05±0.08 | 4.03±0.07 | 4.01±0.07 |
| | 10 | | 4.15±0.09 | 4.08±0.08 | 4.03±0.08 | 4.03±0.07 | 4.03±0.09 | 4.03±0.09 | 4.10±0.10 | 4.03±0.08 | 3.99±0.08 | 4.01±0.08 | 4.03±0.09 | 4.03±0.07 | 4.24±0.10 | 4.21±0.09 | 4.10±0.08 | 4.08±0.08 | 4.03±0.08 | 4.02±0.07 |
| | 25 | | 4.09±0.08 | 4.01±0.08 | 4.01±0.08 | 4.01±0.08 | 4.00±0.09 | 4.02±0.08 | 4.04±0.08 | 3.96±0.07 | 3.90±0.08 | 3.91±0.08 | 3.89±0.08 | 3.92±0.07 | 4.24±0.09 | 4.18±0.09 | 4.10±0.09 | 4.06±0.08 | 4.03±0.08 | 4.01±0.08 |
| | 50 | | 4.06±0.08 | 4.00±0.07 | 4.00±0.07 | 3.98±0.07 | 4.01±0.09 | 3.99±0.08 | 4.04±0.08 | 3.93±0.07 | 3.86±0.09 | 3.83±0.08 | 3.81±0.08 | 3.81±0.10 | 4.24±0.10 | 4.24±0.10 | 4.18±0.09 | 4.11±0.09 | 4.06±0.09 | 4.03±0.08 |
| 3 | 5 | | 4.14±0.09 | 4.08±0.08 | 4.11±0.06 | 4.09±0.11 | 4.11±0.09 | 4.09±0.09 | 4.10±0.09 | 4.10±0.07 | 4.02±0.08 | 4.04±0.08 | 4.15±0.08 | 4.04±0.09 | 4.22±0.09 | 4.10±0.08 | 4.07±0.09 | 3.92±0.07 | 3.90±0.06 | 3.86±0.05 |
| | 10 | | 4.02±0.08 | 4.02±0.08 | 4.00±0.07 | 4.02±0.07 | 4.02±0.07 | 3.99±0.07 | 4.01±0.08 | 3.95±0.07 | 3.92±0.07 | 3.92±0.08 | 3.90±0.07 | 3.90±0.08 | 4.20±0.09 | 4.10±0.08 | 3.98±0.08 | 3.99±0.07 | 4.05±0.08 | 4.03±0.08 |
| | 25 | | 3.96±0.08 | 3.93±0.07 | 3.87±0.07 | 3.87±0.07 | 3.87±0.07 | 3.83±0.07 | 3.88±0.08 | 3.84±0.08 | 3.76±0.07 | 3.67±0.06 | 3.75±0.10 | 3.71±0.09 | 4.24±0.10 | 4.19±0.08 | 4.11±0.09 | 4.06±0.08 | 4.03±0.08 | 4.02±0.08 |
| | 50 | | 3.89±0.08 | 3.88±0.07 | 3.85±0.06 | 3.80±0.09 | 3.82±0.07 | 3.80±0.08 | 3.86±0.08 | 3.77±0.09 | 3.62±0.06 | 3.61±0.08 | 3.59±0.09 | 3.60±0.07 | 4.24±0.10 | 4.24±0.10 | 4.12±0.09 | 4.07±0.08 | 4.04±0.08 | 4.03±0.08 |
| 4 | 5 | | 4.14±0.10 | 4.10±0.08 | 4.08±0.08 | 4.11±0.09 | 4.07±0.05 | 4.09±0.09 | 12.00±3.26 | 4.04±0.07 | 4.06±0.09 | 4.07±0.07 | 4.09±0.08 | 4.07±0.08 | 4.22±0.09 | 4.10±0.08 | 3.97±0.07 | 3.93±0.07 | 3.88±0.07 | 3.85±0.07 |
| | 10 | | 4.01±0.08 | 3.98±0.07 | 3.99±0.07 | 3.99±0.07 | 4.05±0.07 | 4.01±0.06 | 4.03±0.09 | 3.99±0.06 | 3.94±0.06 | 3.94±0.07 | 3.92±0.07 | 3.93±0.08 | 4.20±0.09 | 4.12±0.08 | 4.08±0.10 | 3.94±0.07 | 4.06±0.08 | 4.01±0.08 |
| | 25 | | 3.93±0.09 | 3.91±0.08 | 3.87±0.07 | 3.78±0.06 | 3.84±0.07 | 3.82±0.07 | 3.94±0.07 | 3.85±0.08 | 3.76±0.08 | 3.75±0.08 | 3.77±0.10 | 3.78±0.09 | 4.24±0.09 | 4.18±0.09 | 4.11±0.09 | 4.06±0.08 | 4.04±0.08 | 4.03±0.08 |
| | 50 | | 3.92±0.08 | 3.96±0.07 | 3.78±0.11 | 3.82±0.07 | 3.74±0.10 | 3.70±0.07 | 3.90±0.08 | 3.82±0.06 | 3.71±0.09 | 3.73±0.09 | 3.75±0.08 | 3.77±0.08 | 4.24±0.09 | 4.20±0.10 | 4.12±0.09 | 4.06±0.08 | 4.04±0.08 | 4.03±0.08 |

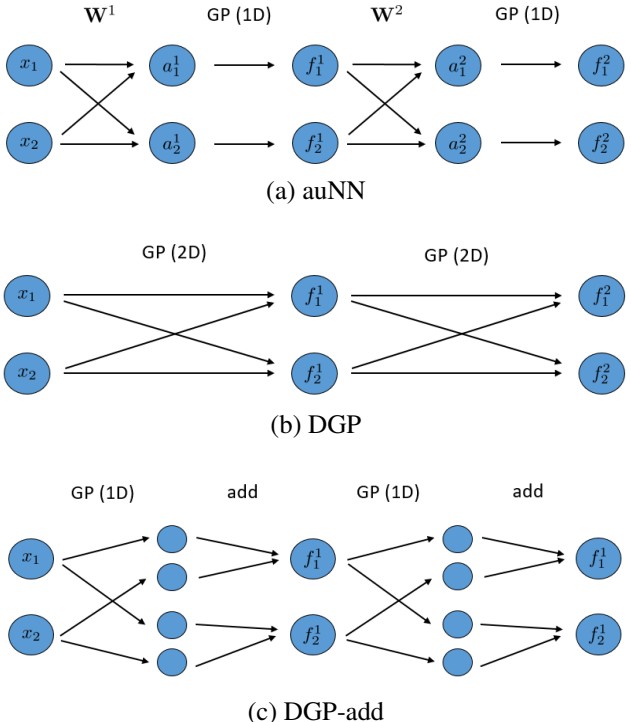

Figure 11: Representation of two hidden layers (with two units per layer) for auNN (a), DGP (b), and DGP-add (c).

## D  COMPUTATIONAL COST SUMMARY

Table 15 shows the training computational complexity for the methods compared in this paper. Moreover, in order to evaluate the computational cost in practice, the table also shows the actual running time for the experiment of Section 3.1. BNN is the fastest algorithm, since it utilizes a factorized Gaussian for the approximate posterior. Although fBNN has the same theoretical complexity, the Spectral Stein Gradient Estimator (Shi et al., 2018) is used to compute the KL divergence gradients. Moreover, a GP prior is specified at function space, for which a GP must be trained as a previous step. DGP and auNN have the same theoretical complexity. In practice, auNN is typically faster because it requires fewer inducing points, recall Section 3.3 and Table 3. The running time in Table 15 is very similar for both because the same amount of inducing points ($M = 10$) is used in this simple experiment.

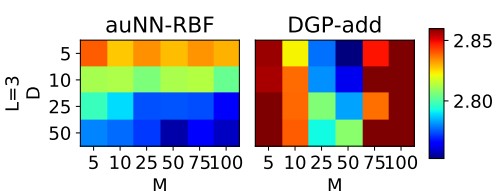

Figure 12: Test NLL on Power dataset for different values of $D$ and $M$ (the lower the better).

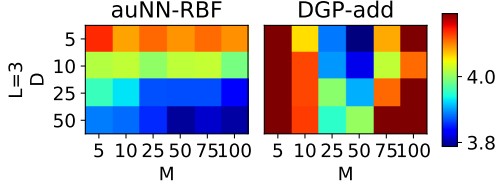

Figure 13: Test RMSE on Power dataset for different values of $D$ and $M$ (the lower the better).

Table 13: Standard error for the results in Table 2. Three random train-test splits are considered.

|  |  | N | D | auNN | | | | | | DGP | | |
|---|---|---|---|---|---|---|---|---|---|---|---|---|
|  |  |  |  | RBF-2 | RBF-3 | RBF-4 | TRI-2 | TRI-3 | TRI-4 | DGP-2 | DGP-3 | DGP-4 |
| Brier | HIGGS | 11M | 28 | 0.0001 | 0.0007 | 0.0008 | 0.0003 | 0.0005 | 0.0009 | 0.0018 | 0.0016 | 0.0006 |
|  | SUSY | 5M | 18 | 0.0005 | 0.0005 | 0.0006 | 0.0005 | 0.0005 | 0.0005 | 0.0011 | 0.0014 | 0.0021 |
| ECE | HIGGS | 11M | 28 | 0.0015 | 0.0020 | 0.0022 | 0.0010 | 0.0035 | 0.0019 | 0.0006 | 0.0004 | 0.0008 |
|  | SUSY | 5M | 18 | 0.0012 | 0.0011 | 0.0014 | 0.0018 | 0.0012 | 0.0014 | 0.0005 | 0.0006 | 0.0008 |

Table 14: Standard error for the results in Table 3. Fifty independent runs are considered.

|  | auNN | | | | | | DGP | | |
|---|---|---|---|---|---|---|---|---|---|
|  | RBF-2 | RBF-3 | RBF-4 | TRI-2 | TRI-3 | TRI-4 | DGP-2 | DGP-3 | DGP-4 |
| HIGGS | 0.0258 | 0.0325 | 0.0371 | 0.0188 | 0.0318 | 0.0378 | 0.0248 | 0.0266 | 0.0269 |
| SUSY | 0.0215 | 0.0274 | 0.0369 | 0.0202 | 0.0258 | 0.0350 | 0.0108 | 0.0126 | 0.0144 |

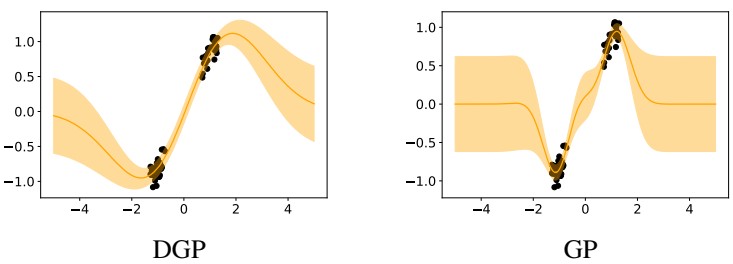

Figure 14: DGP and GP trained on the dataset of Section 3.1. The experimental details are analogous to those in Section 3.1, see Appendix B. Whereas DGP underestimates the uncertainty for in-between data, a simpler GP does provide increased uncertainty in the gap.

Table 15: Training computational cost for the models compared in this paper. The running time (in seconds) corresponds to the mean and one standard error over 10 independent runs of the experiment in Section 3.1. More details in Appendix D.

|  | BNN | fBNN | DGP | auNN |
|---|---|---|---|---|
| Running time (s) | $15.21 \pm 0.78$ | $51.92 \pm 1.07$ | $22.37 \pm 0.97$ | $21.16 \pm 0.89$ |
| Complexity | $\mathcal{O}(N \sum_i D^i D^{i+1})$ | $\mathcal{O}(N \sum_i D^i D^{i+1})$ | $\mathcal{O}(NM^2 \sum_i D^i)$ | $\mathcal{O}(NM^2 \sum_i D^i)$ |

