# OpenReview forum: "Activation-level uncertainty in deep neural networks"
_ICLR.cc/2021/Conference — ICLR 2021 Poster_

### Official Review · AnonReviewer1 · 2020-10-27
**Activation-level uncertainty is an interesting method to model uncertainty in the function space.**

**Rating:** 7
**Confidence:** 3

**Review:**


In this paper, the authors contribute to the recent trend of replacing uncertainty in the weight space in favor of uncertainty in the function space. Their main contribution, of implementing activation-level uncertainty, is a valuable one and will inspire future research.

Overall, this paper is well-motivated, clear-to-read, novel, and interesting. The mathematical reasoning is sound and the Figures are of excellent quality: in particular, Figure 2 is a very good illustration of the paper's core concepts. Figure 6 is also clear, thought-provoking, and informative.

To improve the work, the authors could devote more time to discussing the limitations of the proposed method. In particular, the authors make a point of the scalability: however, the datasets used in the network, while having many training samples, are quite small in their dimensionality (D=28 for HIGGS and 18 for SUSY). While the work compares extensively against deep GPs, a broader set of comparisons would improve the paper. Work such as 'Cyclical Stochastic Gradient MCMC for Bayesian Deep Learning' shows that Bayesian inference can be scaled to work on ImageNet, using resnet-50 — although this work is not cited in the paper, it does show that existing approaches can scale to large networks and large datasets. Finally, the computational complexity of the proposed method should be outlined more explicitly — although there is some content in a footnote, a table comparing different models and their requirements would be of interest.

In terms of the modeling, I would also appreciate some discussion of the 'independence between units' assumption in the probabilistic model. The assumption that the uncertainty of a given unit depends only on the value of its activation seems to me to be a fairly strong one, so some more explicit discussions of the possible consequences of this choice would improve the work.

One structural change I would recommend is to move the 'related work' section forward: it currently precedes the conclusion, whereas I feel it would be better earlier in the manuscript to set the scene more thoroughly.

There are a couple of more minor points to address:

* I find the 'Introduction' section of the work to be poorly-researched. All of the underlying motivations of modeling uncertainty have references to deep-learning papers from the last 5 years or so. It has the unfortunate effect of implying that people in these fields had never thought about these issues prior to the current deep learning boom. Using the cited papers would be fine if they were phrased more like "deep learning has applied uncertainty estimation to the following critical applications", but if the authors want to support the statements they've actually written, they should do more research. A possible starting point would be Bishop, §1.5.3.
* The authors refer often to 'reversing' when they mean 'reverting' (to revert is to return to a previous condition, which is exactly the meaning the authors want to convey).
* The paper https://openreview.net/pdf?id=B1l08oAct7 could also be referenced and discussed.

---

> ### Author Response · Authors · 2020-11-23
> **Author response to AnonReviewer1**
>
> We thank the reviewer for their valuable feedback, which has helped us to improve the work.
> We reply to each question/comment below.
>
> **1. More discussion on the limitations of the proposed method.** (Third paragraph).
>
> Although the maximum dimensionality of the datasets used in the paper is $D=28$ (HIGGS dataset), this is not a theoretical limitation of the approach.
> In order to deal with a larger input dimensionality, one would only need to consider higher dimensional weights in the first layer.
> However, we do believe that efficiently scaling up to larger network architectures (i.e. very wide and/or deep networks) could be more challenging, since the computational cost of auNN depends on $D^1+\dots+D^L$.
> This is discussed now in the future work (Section 5), where we envision a simpler activation-level uncertainty model for these cases.
> As for the computational complexity of auNN, we agree that a more explicit comparison could be of interest for the reader.
> Therefore, we have added a new appendix (Appendix D, in particular Table 15) summarizing the computational complexity and the running time for all the compared methods.
>
>
> **2. On the "independence between units" assumption.** (Fourth paragraph).
>
> Notice that we are referring to conditional independence given the values of the activations (i.e. in turn, given the values of the previous layer).
> This is conceptually natural, since the different units are given by different weights and treated independently.
> Indeed, assuming independence between the different units of the same layer is a standard approach when defining probabilistic deep models.
> See, for instance, the analogous assumption in DGPs (first two sentences of Section 2.2, which refers to general DGPs, https://arxiv.org/abs/1705.08933).
>
> **3. Location of the "related work" section.** (Fifth paragraph).
>
> We lean to keep the related work section after the experimental one, since our experience is that this is better understood after knowing how the proposed method works both theoretically and empirically.
> However, we agree that a general overview of the related work can be beneficial before going into the details of the approach.
> Therefore, although we have decided to keep this section where it was originally, we have added a reference to it at the end of the introduction.
>
> **4. Minor aspects.** (Last paragraph).
>
> We have added more classical references to the introduction, and partially re-written the first paragraph. We believe that now it is more clear that, as pointed out by the reviewer, uncertainty estimation in DNNs is not a new field.
>
> Thank you, we agree that "revert" is a more appropriate term. We have adopted it in the revised version.
>
> We have included the suggested paper in the second paragraph of the related work section.

---

### Official Review · AnonReviewer3 · 2020-10-28
**Clearly written and well motivated**

**Rating:** 8
**Confidence:** 4

**Review:**

The authors capture uncertainty in deep networks by employing stochastic activation functions but deterministic network weights.  They place a Gaussian process (GP) prior over activation functions, propose a customized GP kernel, and demonstrate that activation uncertainty leads to both more sensible extrapolations and better-calibrated interpolation uncertainties when compared to standard Bayesian neural networks (BNNs)  with weight uncertainty and deterministic activations.

The observation that BNNs have too many parameters for meaningful posterior inference is widely acknowledged; modeling activation uncertainty instead of weight uncertainty is an interesting approach for alleviating this issue.  While GP based activations have been considered before, the paper does a particularly good job of executing on the idea and clearly demonstrating its advantages. The experiments are well thought out and nicely illustrate the benefits afforded by the paper's contributions.  Overall, I  enjoyed reading this paper and only have minor quibbles.

1. As with any GP based approach computational scalability is a concern. While it is reassuring that only a modest number of inducing points are needed, the complexity of one epoch: $O(NM^2(D_1 …+ D_L))$ suggests that it would be challenging to scale auNN to deeper / wider networks. Section 3.4 demonstrates scalability to large data but it does not demonstrate scalability to large architectures. It would be good to include a discussion of this issue and how the authors envision scaling the approach to more standard deep learning architectures. Table 3 should include timing numbers for both standard variational BNNs and functional BNNs (even if they are not competitive in terms of performance). Having these numbers would help guide follow up work interested in replacing GPs with (potentially more scalable) parametric function approximations.
2. The choice of the GP kernel seems to have a large effect on the learned activations (Figure 8). It would be interesting to consider uncertainty over GP kernels as opposed to a-priori fixing all activations to be drawn from GPs with the same kernel and potentially allow different layers or different units in a layer to prefer activations drawn from GPs with different kernels.
3. I found the sentences (section 3.1, page 6, last paragraph) highlighting differences between a deepGP with an additive kernel and auNN confusing. I think the point being made is that in auNN different nodes in layer $l+1$ use a *shared* set of distinct $D_l$ functions, with each node weighting these functions differently. While in DGP there are no shared functions. From the text it almost sounds like that in auNN functions in layer $l$  (all $D_l$ of them) are the same, which doesn’t seem to be the case.
4. The observation that many BNN inference techniques struggle with appropriate “gap” uncertainty was concurrently made by Foong et al., and Yao et al., https://arxiv.org/pdf/1906.09686.pdf

---

> ### Author Response · Authors · 2020-11-23
> **Author response to AnonReviewer3**
>
> We thank the reviewer for their positive review and their interesting comments, which have contributed to improve the manuscript.
> We reply to each question/comment below.
>
> **1. More discussion on scalability.** (First point in the enumeration).
>
> Regarding the scalability to larger architectures (i.e. wider and/or deeper networks), we have included a discussion in the future work section (Section 5), where we envision a simpler activation-level uncertainty model for these cases.
> As for the running time of BNN/fBNN, we agree that it is interesting to provide the timing numbers for all the methods.
> Since *AnonReviewer1* also asks about the computational cost of the different methods, we have included a new appendix (Appendix D) devoted to these aspects.
> Specifically, in Table 15, we provide the running time of all the methods (BNN, fBNN, DGP, auNN) for the experiment of Section 3.1.
> The performance of the different methods is explained in the text of Appendix D. Notice that we have not just expanded Table 3 because fBNN is too slow in these very large datasets.
>
>
> **2. Uncertainty over GP kernels.** (Second point in the enumeration).
>
> A detailed study on the properties induced by each type of kernel is one of the main lines of future work.
> The idea of letting the algorithm learn the optimal kernel is definitely an interesting one.
> We have added this to the discussion on future work (Section 5).
>
> **3. Clarifying the sentences on the differences between DGP-add and auNN.** (Third point in the enumeration).
>
> Yes, we totally agree on this, in the original version it could be understood that all the functions are actually the same.
> We have rewritten this paragraph clarifying this.
> We have also added more details on the similarity between DGP-add and auNN, as requested by *AnonReviewer4*.
>
> **4. Other works dealing with "gap" uncertainty.** (Fourth point in the enumeration).
>
> We had cited the arxiv work by Foong et al.
> That work has been accepted for NeurIPS 2020, so we have updated the reference.
> We were not aware of the work by Yao et al., thank you. We have also added that reference in the introduction.

---

### Official Review · AnonReviewer2 · 2020-10-28

**Rating:** 6
**Confidence:** 5

**Review:**

Either putting the uncertainty on the weights (e.g., Bayes by BP) or on the activation (e.g., fast dropout or variants of natural-parameter networks [2,3] or Bayesian dark knowledge [4]) or both [1] have been investigated before. The idea of moving the uncertainty from the weight to the activation function is not new. One could argue that VAE-style parameterization or local reparameterization trick is also a kind of methods that put uncertainty in the activation function. In fact the proposed method does involve the reprarameterization trick in each layer as shown in Eq. 7.

The hypothesis that introducing stochasticity in the activation functions is effective in handling uncertainty estimation has also been investigated before [1,2,3].

In terms of uncertainty estimation OOD data, note that BDK [4] and natural-parameter networks (NPN) [1] also claimed satisfactory performance in uncertainty estimation OOD data (see figures in both papers on the toy regression dataset). Therefore it would be interesting to make proper comparison.

This work is heavily built on the DGP work with help from the reparameterization trick. The author mentioned their differences at the end of Sec. 2, emphasizing that DGP define the function on D^{l-1} dimensions while the proposed method is on 1D. Looking from another perspective, the proposed method combined with the previous linear layer can also be seen as ‘defining the function on D^{l-1} dimensions’. In this sense, I would guess that the different is really on the factorization, i.e., how they factorize the linear plus nonlinear operations.

Given that this is built on GP, a complexity analysis on running time (both for training and inference) is provided. I am wondering what is the scale for M, i.e., the number of inducing points, for the method to work properly. Also, it would be good to provide and compare the actual running time.

Experiments are either on toy datasets or two very small data UCI datasets. It would be more convincing to evaluate on larger/higher-dimentional data.

In Figure 3, it is mentioned that BNN underestimate uncertainty between two clusters. I was wondering whether this is true for all types of BNNs. As an example, see Figure 1 of [1], where areas with fewer points do result in higher uncertainty from the PBP and NPN models (which also handles uncertainty in the activation). This comes back to my concern on the experiments. It would also have been stronger if more baselines are included to make a more convincing case.

Another important experiment that is missing is to include DGP in Figure 3. Would DGP perform similarly or even better than auNN?

Another problem with the toy dataset in Figure 3 is that: it is hard to say which way of extrapolation is ‘more correct’. Should the model extrapolate like BNN does or like fBNN does. Given that the input is in a continuous space and that inside each cluster the output is indeed increasing with the input, I would say extrapolating as BNN does also makes sense.

The presentation may be improved. For example, it would be helpful to make it clear what U^l denotes from the beginning. It would also be good to better structure Sec. 2.

[1] Natural-Parameter Networks: A Class of Probabilistic Neural Networks, NIPS 2016
[2] Feed-forward Propagation in Probabilistic Neural Networks with Categorical and Max Layers, ICLR 2018
[3] Sampling-free Epistemic Uncertainty Estimation Using Approximated Variance Propagation, CVPR 2019
[4] Bayesian Dark Knowledge, NIPS 2015

---------
After rebuttal

After the discussion phase, I agree that with R3 that in cases such as with periodic patterns, auNN would be a better way to incorporate such knowledge. In this case, the paper would be stronger if related experiments on such patterns are included, to highlight what the key difference between auNN and typical BNN.

Gap uncertainty is indeed a real problem in BNN, and I am glad that auNN performs well on this, this is also why I would like to see more comparison to other baselines (e.g., NPN) with good performance on gap uncertainty, to better gauge its improvement/capability (e.g., see https://arxiv.org/pdf/1611.00448.pdf Figure 1(right) where there is also a gap in the middle, although not as large as the example from the auNN paper).

If this paper is accepted, it would be great if the author could include R3’s summary during the discussion into the final version (see below) as well as proper BNN baselines, which I think will be quite helpful in positioning the current work:

‘Typical BNNs use stochastic weights and deterministic activations, while the authors’ model (auNN) uses deterministic weights and stochastic activations. You are definitely correct in asserting that uncertainty in weights manifests as uncertainty in activation functions. Directly modeling activation function uncertainty however provides distinct advantages, has not been studied carefully before.’

Experiments mentioned in the first paragraph would also be a good addition to make the paper’s point.

---

> ### Author Response · Authors · 2020-11-23
> **Author response to AnonReviewer2 (1/2)**
>
> We thank the reviewer for their insightful feedback. We reply to each question/comment below.
>
> **1. Uncertainty in the activation has already been studied.** (First and second paragraphs in the review).
>
> We have added a new paragraph in the related work section explaining how our approach is different from the methods based on Natural-Parameter Networks (i.e. [1,2,3] in the review).
> The approach is conceptually different.
> In NPN [1], the non-linear transformations are given by *deterministic* activation functions.
> See, for instance, Section 2.3 of the paper (https://arxiv.org/pdf/1611.00448.pdf), where $v(\cdot)$ denotes a determinisitc activation function (indeed, in the third paragraph they mention the popular cases of $tanh$ and $\mathrm{ReLu}$).
> See also the following sentence in the fourth paragraph of the introduction "Input distributions go through layers of linear and nonlinear transformation
> *deterministically* before producing distributions to match the target output distributions").
> It is true that the output of the activation function is represented by a probability distribution in NPNs and related methods. But this is because the input to the activation function is a probability distribution, not because the activation function is treated stochastically.
> However, auNN models the activations themselves stochastically with a GP prior.
> As for the work "Bayesian Dark Knowledge" (BDK, i.e. [4] in the review), the idea there is to train a standard (i.e. no Bayesian) neural network to distill a Bayesian neural network (which has the prior defined in weight-space).
> From that perspective, the connection of BDK with stochastic activations (and hence with our proposed method, auNN) is not entirely clear to us.
>
> **2. Other methods have also claimed satisfactory performance in uncertainty estimation OOD data.** (Third paragraph).
>
> The reviewer refers here to the uncertainty estimation for OOD data.
> In the paper we refer to the actual extrapolation to OOD data (i.e. the predictive *mean*, not the uncertainty).
> In fact, in terms of uncertainty, notice that BNN is also obtaining satisfactory performance (i.e. the uncertainty increases as we move far from the training data).
> Once this has been clarified, it still remains the question whether the actual extrapolation to OOD data (i.e. the predictive mean) is good or not.
> Actually, this is something that the reviewer also asks.
> Please see our reply to that specific question (point 8 below).
>
> **3. Difference between DGP and auNN.** (Fourth paragraph).
>
> Of course auNN units can be seen as functions defined on $D^{l-1}$ dimensions if the 1D GP is combined with the previous linear layer.
> In fact, Figure 2a in the paper shows the differences between DGP and auNN units when both are viewed as functions on $D^{l-1}$ dimensions ($D^{l-1}=2$ in this case).
> As can be seen there, the fact that auNN applies deterministic weights plus a 1D GP makes its units simpler (i.e. defined along the direction of the linear projection).
> This results in the different empirical behavior shown in Section 3.3 (Figure 6b-c), with auNN needing fewer inducing points and benefiting more from deeper architectures.
> Also notice that, whereas the inducing points are $D^{l-1}$-dimensional in DGP (just like the input), they are 1-dimensional in auNN, and this is precisely due to the previous linear projection.
>
> **4. Scale of $M$ and actual running time.** (Fifth paragraph).
>
> Different values of $M$ (and also of $D$, the number of hidden units per layer) are analyzed in Figure 6b-c for auNN (and also for DGP).
> These figures show that, in general, auNN achieves its best performance for $M$ around $M=25$, and larger values of $M$ do not produce better results.
> This is as theoretically expected, since $25$ inducing points are usually enough to model a simple 1D activation function.
> Interestingly, as explained in Section 3.3, this is an advantage of auNN over DGP, since the latter requires more inducing points.
> In the rest of experiments, we use for auNN either $M=25$ (in SUSY and HIGGS, which are larger and more complex datasets) or $M=10$ (in the UCI datasets and in the toy one).
> As for the actual running time of auNN, the reviewer is referred to Table 3. There, the training time of auNN is shown to be lower than that of DGP (precisely because it requires fewer inducing points).
> For a more comprehensive outline on the computational cost of all the compared methods, please see the new Appendix D (in particular, Table 15).

---

> > ### Author Response · Authors · 2020-11-23
> > **Author response to AnonReviewer2 (2/2)**
> >
> > **5. More sophisticated datasets needed.** (Sixth paragraph).
> >
> > Please note that we do evaluate our method on large datasets containing up to eleven million data instances. As a matter of fact, the detailed list of datasets considered in our experiments is:
> >
> > * One toy dataset in Section 3.1.
> > * Six UCI datasets (both with gap and standard splits) in Section 3.2. These are Boston, Concrete, Energy, Naval, Wine, and Yacht (see e.g. Table 4). Among these, the maximum size is $N=11934$ (Naval), and the maximum dimensionality is $D=26$ (Naval).
> > * Two more UCI datasets are used in Section 3.3. These are Kin8 and Power.
> > * Two large-scale particle physics datasets are used in Section 3.4. These are HIGGS ($N=1.1\times 10^7$, $D = 28$) and SUSY ($N=5\times 10^6$, $D = 18$).
> >
> > **6. There are already methods that do not underestimate uncertainty for data in-between clusters.** (Seventh paragraph).
> >
> > We agree that there are other methods which can provide satisfactory uncertainty for in-between data.
> > Think, for instance, of a simple Gaussian Process.
> > Indeed, this is related to the next question, see point 7 below.
> > In Figure 3 we are comparing to BNN (trained with Bayes by Backprop) and the recent fBNN because their formulations are the most related to ours, recall the second and third paragraphs in the introduction.
> > BNN introduces the stochasticity in the space of weights.
> > As for fBNN, although the prior is specified in function-space, the approximate posterior is still defined on weight-space (see Section 3.1 in their paper).
> > We propose a different alternative: to introduce the stochasticity in the activation functions, and treat the weights deterministically.
> > Therefore, the goal of the figure is to illustrate that this novel alternative alleviates an issue shared by the two other approaches.
> > From Figure 1 in [1], it is difficult to judge how PBP and NPN would estimate uncertainty for in-between data, since the setting in that figure is fairly different. However, these methods are less related to auNN and older than fBNN.
> > Finally, for more details on the estimation of uncertainty for in-between data, the reviewer is referred to https://arxiv.org/abs/1909.00719 (this work has been recently accepted at NeurIPS 2020; our reference has been updated accordingly).
> >
> > **7. DGP in Figure 3.** (Eight paragraph).
> >
> > As explained in the previous question, the goal of Figure 3 is to compare three related approaches to model uncertainty in neural networks (i.e. BNN, fBNN and auNN).
> > DGP has not been included because it is not usually considered a type of neural network (since it does not combine linear and non-linear transformations).
> > However, as suggested by the reviewer and driven by curiosity, we have run a DGP on this dataset.
> > Just like auNN, it uses $L=2$ (one hidden layer), $D=25$ hidden units, $M=10$ inducing points, and is trained during $5000$ epochs.
> > The results are in Figure 14 in the appendix.
> > We observe that DGP underestimates uncertainty for data in-between the two clusters of training points.
> > Figure 14 also shows a single GP, which does provide better uncertainty estimations.
> > This is an interesting example showing that, in general, DGPs are not always better than GPs.
> >
> >
> > **8. What is "good extrapolation" in Figure 3?** (Ninth paragraph).
> >
> > We agree that, in principle, there is no reason to consider one extrapolation "more sensible" than the other.
> > What we really mean here is that, by having a prior at function or activation level (i.e. as in fBNN or auNN), one is able to guide the extrapolation to OOD data, for instance, by reverting to the empirical mean.
> > However, by specifying the prior in the complex and highly dimensional space of weights, we loss such mechanism to guide the extrapolation to OOD data.
> > We have made this clear in the revised manuscript, specially in the introduction and in Section 3.1.
> >
> > **9. Presentation.** (Tenth paragraph).
> >
> > To improve the presentation of Section 2, we have included two more sub-parts ("model specification" and "variational inference").
> > We have also clarified what $\mathbf{U}^l$ denotes when it first appears.

---

### Official Review · AnonReviewer4 · 2020-10-29
**the technique is interesting but not novel enough**

**Rating:** 6
**Confidence:** 4

**Review:**

The paper proposes a variant of BNN with stochasticity added to the activation layer (auNN). It's proposed to solve the underestimation of uncertainty for instances located in-between two clusters of training points. It also shows comparison to DGP and indicates that auNN is better than DGP by requiring less inducing points and better suited for deep architectures. There is a strong connection between auNN with DGP which is discussed in the later part of the paper which I appreciate.

However, I think there are still a few things needed to be more explicitly stated: In page 6, "Whereas auNN considers combinations of the same Dl 1D functions to define the (l + 1)-th layer from the l-th one, DGP-add describes Dl different 1D functions for each unit in the (l + 1)-th layer. " This is not quite clear, why auNN combines the same functions while DGP combines different. From figure 1, it seems that the only difference between auNN and DGP is whether f1,f2,f3 are summed up into a scalar a or not. Which part is different or same? I think it might need a bit math to demonstrate that and it would be helpful to draw figures for multiple nodes in the lth layer rather than just one. Cuz from the description under eq1 and figure 1, my understanding is f1, f2 and f3 in the l-1th layer are combined together with w to get a scalar a. Then a 1D GP is constructed from this scalar a to generate a scalar f for the lth layer. When we want another f in the lth layer, we just use another w to linearly combine f1,f2 and f3 (l-1th layer) and generate f using another GP from a. While in an ordinary DGP, fd in the lth layer is from a kernel taking f1,f2,f3 (l-1th layer) all together. So presumably, fd in auNN is from different GP functions with different kernels while fd in DGP is from the same GP with the same kernel. I just think this connection/distinction is a bit confusing to me. A better way to clarify it is using mathematical equations to express the unit in auNN and DGP respectively, including what the kernel looks like, what the expression for fd is, etc. There are such definitions for auNN but not DGP, and conceptual-level description is not clear enough.

Moreover, although the paper emphasized the difference between auNN and DGP, I still think auNN is a specific/simplified version of DGP. Here is the reason: in auNN, f1 is only dependent on a1 and f2 is only dependent on a2. This is saying, the joint distribution of (f1,f2) has a block diagonal covariance and the blocks are different for each d. In DGP, f1 and f2 are usually considered to be independent to each other, thus the joint distribution of (f1,f2) also has a block diagonal covariance but the blocks are the same. But people can make DGP more general by allowing different blocks (kernel functions) for f1 and f2. Another difference is the input to the kernel (It would be appreciated too if the authors can explicitly write out the kernel definitions for auNN and DGP for easy understanding). For DGP, the input to fd (lth layer) is a D-dimensional vector (f1,f2,...,fD) in the l-1th layer; different fd share the same inputs. For auNN, the input to fd (lth layer) is a scalar ad=wd*(f1,f2,...,fD), and different fd have different inputs since wd is different. This part is more like different designs of the kernel. If so, then the inference method for DGP would be just applicable to auNN (adding the optimization for w).

In sum, this design of kernel and independent fd is interesting but the overall novelty is not strong enough from both the methodology perspective and inference perspective.

A few other things:
1. In the intro, it says "First, it has been recently shown that BNNs with well-established inference methods ... underestimate the predictive uncertainty for instances located in-between two clusters of training points. Second, BNNs do not extrapolate sensibly to out-of-distribution (OOD) data. Both issues are illustrated graphically in Figure 3." I don't see both issues in Figure 3. Or more precisely speaking, what's the difference between these two points?

2. Define d when it first shows up in the paper.

3. Define the dimensions for mud, Kd.

4. It shows that auNN requires less inducing points but the number of parameters is actually more than DGP due to w. It would be helpful to have such a discussion too. Since people know that the optimization for DGP could be tricky even with two parameters (e.g., length scale and marginal variance in RBF) in the kernel. Now the kernel has more parameters, how it would behave during the optimization and at the optimum? I think the larger variance of auNN across layers might be due to w, so comparing auNN with DGP-add might a bit unfair.

----------------------------------------------------
I raise my score to 6 after the rebuttal given the clarification the author added to answer my confusion.

---

> ### Author Response · Authors · 2020-11-23
> **Author response to AnonReviewer4 (1/2)**
>
> Thank you for thoroughly reading our paper and bringing up interesting questions to discuss.
> We reply to each question/comment below.
>
> **1. Are auNN and DGP-add using "the *same* or *different* functions" to define the $(l+1)$-th layer from the $l$-th one?** (Second paragraph of the review).
>
> The understanding of the reviewer expressed in the second half of the paragraph (i.e. from "Cuz from the description under eq1 and figure 1, my understanding is...") is correct.
> We believe that the confusion here is because we are referring to DGP-add (i.e. DGP with additive kernel), and we may not have explained enough the particular connection between DGP-add and auNN.
> To make this clear, we have included a figure for DGP-add showing all the units for all the layers (and we have also included all the units for all the layers for DGP and auNN, as suggested by the reviewer). This new figure is Figure 11 in Appendix C.
>
> Based on these figures, let us clearly explain the connection between auNN and DGP-add.
> Indeed, notice that the model for *$a^{l+1}|a^l$ in auNN* is very similar to that of *$f^{l+1}|f^l$ in DGP-add*. Specifically, in both cases, the input ($a^l$ in auNN, $f^l$ in DGP-add) goes through 1D GPs and then these are aggregated (linear combination through $\mathbf{W}$ in auNN, summation in DGP-add) to yield the output ($a^{l+1}$ in auNN, $f^{l+1}$ in DGP-add).
>
> However, as mentioned in the paper, there exists a key difference.
> In auNN, all the nodes in the $(l+1)$-th layer (i.e. $a_i^{l+1}$ for $i=1,2$) aggregate a *shared* set of distinct functions (namely, $f^l_i$ for $i=1,2$), each node using its own weights to aggregate them.
> While in DGP-add, there is not such shared set of functions, and each node in the $(l+1)$-th layer (i.e. $f_i^{l+1}$ for $i=1,2$) aggregates a different set of GP realizations (the unlabelled blue nodes in Figure 11c).
>
> To address this, in the revised manuscript we have rewritten the last paragraph of Section 3.3 to make clearer the connection between auNN and DGP-add. We have also referred there Figure 11 in Appendix C to illustrate it visually.
>
> We also point the reviewer to the third item in the enumeration by AnonReviewer3. They also find this sentence confusing and rephrase it in a way that is enlightening in our opinion.
>
> **2. auNN is not novel enough when compared to DGP.** (Third and fourth paragraphs of the review).
>
> We agree that auNN is similar to a DGP model in which the weights $\mathbf{w}_d^l$ can be seen as kernel parameters.
> To be more specific, let us focus on auNN with a RBF kernel. This model is very similar to a DGP in which each unit $f^{l+1}_d$ is given by a GP on $\mathbf{f}^l$ with the kernel $k(\mathbf{x},\mathbf{y})=\gamma\cdot\exp\left(-\frac{||(\mathbf{w}_d^{l+1})^\intercal\mathbf{x}-(\mathbf{w}_d^{l+1})^\intercal\mathbf{y}||^2}{2\ell^2}\right)$, where now $\mathbf{w}_d^{l+1}$ are kernel parameters.
> As the reviewer summarizes in the fourth paragraph, this is a specific kernel design, and each unit $f^{l+1}_d$ uses a different kernel (since the kernel depends on $\mathbf{w}_d^{l+1}$).
>
> However, *auNN models the weights $\mathbf{w}_d^{l+1}$ with a previous linear layer*, which makes it a qualitatively different model.
> For instance, consider the dimensionality of the inducing points.
> In a DGP with the ($\mathbf{w}_d^{l+1}$)-dependant kernel, the inducing points would be $D^l$-dimensional (just like the input $\mathbf{f}^l$).
> In auNN, the inducing points are 1-dimensional (just like the input $a_d^{l+1}$), which significantly reduces the amount of parameters to estimate.
> This is is precisely because in auNN the projection through $\mathbf{w}^{l+1}_d$ is modelled with a previous linear layer, and it is not part of the kernel.
> This also implies that the inference method for DGP is not just applicable without modifications to auNN.
>
> Moreover, this different treatment of the parameters $\{\mathbf{w}_d^l\}$ allows for establishing an interesting connection with neural networks, since auNN can be seen as a neural network that models the uncertainty in the space of activations.
> Further, this connection with neural networks motivated the use of the TRI kernel, which models piecewise linear functions similar to the ReLu (which is known to be a popular activation function for neural networks).

---

> > ### Author Response · Authors · 2020-11-23
> > **Author response to AnonReviewer4 (2/2)**
> >
> > **3. Clarification on how two aspects mentioned in the introduction are different and how they are illustrated in Figure 3.** (First point in the enumeration).
> >
> > We are talking about two things here: 1) the underestimation of uncertainty for in-between data and 2) the extrapolation to out-of-distribution (OOD) data.
> >
> > Let us focus on Figure 3 to explain both points.
> > Let us start with the former (underestimation of uncertainty for in-between data).
> > Observe that the training dataset has two clusters of training points, one around $x=-1$ and the other one around $x=+1$, and there is no observed data in between (i.e. in the interval $x\in (-1,1)$).
> > Therefore, it would be desirable that the predictive uncertainty in this "gap" were higher than around $x=\pm 1$.
> > Figure 3 shows that BNN and fBNN do not achieve this, whereas auNN does (notice the wider shaded area for $x\in(-1,1)$ in auNN).
> > As explained in the paper, the underestimation of uncertainty for in-between data has been thoroughly analyzed in https://arxiv.org/abs/1909.00719 (this has been recently accepted at NeurIPS 2020, and the reference in our paper has been updated accordingly).
> > The authors there observe that specifying the approximate posterior in the space of weights is related to the underestimation of in-between uncertainty.
> > Interestingly, this is overcome by auNN.
> >
> > Let us focus now on the latter (extrapolation to OOD data).
> > Whereas BNN goes to $+\infty$ when $x\to +\infty$ (and $-\infty$ when $x\to -\infty$), fBNN and auNN revert to the empirical mean of the observed data when $x\to\pm\infty$ (notice that the empirical mean of the observed data is around zero, since one cluster of samples has y-value around $+1$ and the other one around $-1$).
> > However, as pointed out by AnonReviewer2, we acknowledge that, in principle, there is no reason to consider one extrapolation "more sensible" than the other.
> > What we really mean here is that, by having a prior at function or activation level (i.e. as in fBNN or auNN), one is able to guide the extrapolation to OOD data, for instance, by reverting to the empirical mean.
> > We have made this clear in the revised manuscript, specially in the introduction and in Section 3.1.
> >
> >
> >
> >
> > **4. Including some definitions.** (Second and third points in the enumeration).
> >
> > In Figure 1, the subscript $d$ and the superscript $l$ refer to the $d$-th unit in the $l$-th layer, respectively. We have explicitly included this in the caption.
> > In Section 2, $\boldsymbol{\mu}_d^l$ is a $N$-dimensional vector and $\mathbf{K}_d^l$ is a $N\times N$ matrix. We have explicitly specified these dimensions when they first appear.
> >
> > **5. In auNN there are more parameters to optimize than in DGP, due to the weights $\mathbf{w}$.** (Fourth point in the enumeration).
> >
> > It is true that weights $\mathbf{w}$ are new parameters which need to be optimized.
> > However, it is not only that auNN requires fewer inducing points, but also the fact that these (their locations) are 1-dimensional in auNN, whereas they are $D^{l}$-dimensional in DGP. This additionally reduces the amount of parameters to be optimized by a factor of $D^l$ in each layer.
> > In practice, the proposed initialization for the weights (the Glorot uniform initializer, recall Appendix A, which is based on the analogy between auNN and neural networks) provides a stable convergence for auNN.

---

### Decision · Program_Chairs · 2021-01-07
**Final Decision**

**Decision:**

Accept (Poster)

**Comment:**

This paper presents  a new approach to model uncertainty in DNNs, based on deterministic weights and simple stochastic non-linearities, where the stochasticity is encoded via a GP prior with a triangular kernel inspired by ReLu. The empirical results are promising. The comments were properly addressed. Overall, a good paper.